# Discovery proteomics in aging human skeletal muscle finds change in spliceosome, immunity, proteostasis and mitochondria

Ceereena Ubaida-Mohien[1], Alexey Lyashkov[1], Marta Gonzalez-Freire[1], Ravi Tharakan[1], Michelle Shardell[1], Ruin Moaddel[1], Richard D Semba[2], Chee W Chia[1], Myriam Gorospe[1], Ranjan Sen[1], Luigi Ferrucci[1]*

[1]Intramural Research Program, National Institute on Aging, National Institutes of Health, Baltimore, United States; [2]Johns Hopkins Medical Institute, Baltimore, United States

**Abstract** A decline of skeletal muscle strength with aging is a primary cause of mobility loss and frailty in older persons, but the molecular mechanisms of such decline are not understood. Here, we performed quantitative proteomic analysis from skeletal muscle collected from 58 healthy persons aged 20 to 87 years. In muscle from older persons, ribosomal proteins and proteins related to energetic metabolism, including those related to the TCA cycle, mitochondria respiration, and glycolysis, were underrepresented, while proteins implicated in innate and adaptive immunity, proteostasis, and alternative splicing were overrepresented. Consistent with reports in animal models, older human muscle was characterized by deranged energetic metabolism, a pro-inflammatory environment and increased proteolysis. Changes in alternative splicing with aging were confirmed by RNA-seq analysis. We propose that changes in the splicing machinery enables muscle cells to respond to a rise in damage with aging.
DOI: https://doi.org/10.7554/eLife.49874.001

*For correspondence:
FerrucciLu@grc.nia.nih.gov

**Competing interests:** The authors declare that no competing interests exist.

## Introduction

One of the most striking phenotypes of aging is the decline of skeletal muscle strength, which occurs in all aging individuals and contributes to the impairment of lower extremity performance and loss of mobility (*Cruz-Jentoft et al., 2010*; *Studenski et al., 2014*; *Cesari et al., 2015*). The magnitude of decline in strength is higher than that expected from the loss of muscle mass, suggesting that the contractile capacity of each unit of muscle mass is progressively lower with aging. The reasons for such a decline of contractile capacity are unclear, and several hypotheses have been proposed (*Moore et al., 2014*). Studies conducted in humans by $^{31}$P magnetic resonance spectroscopy as well as 'ex vivo' respirometry have shown that skeletal muscle oxidative capacity declines with aging and such decline affects mobility performance (*Gianni et al., 2004*; *Hepple, 2016*; *Zane et al., 2017*; *Gonzalez-Freire et al., 2018*). Ample evidence from animal models, and more limited evidence from human studies also suggest that aging causes progressive muscle denervation, with enlargement of the motor units and degeneration of the neuromuscular junction, but whether these changes account for the change of contractile performance of human muscle with aging has not been studied (*Wang et al., 2005*; *Messi et al., 2016*; *Delbono, 2003*; *Spendiff et al., 2016*; *Gonzalez-Freire et al., 2014*).

Currently, no treatment is available to prevent or delay the decline of muscle strength and function with aging. Thus, understanding the mechanisms driving the decline in muscle contractile

**eLife digest** As humans age, their muscles become weaker, making it increasingly harder for them to move, a condition known as sarcopenia. Analyzing old muscles in other animals revealed that they produce energy inefficiently, they destroy more proteins than younger muscles, and they have high levels of molecules that cause inflammation. These characteristics may be involved in causing muscle weakness.

Proteomics is the study of proteins, the molecules that play many roles in keeping the body working: for example, they accelerate chemical reactions, participate in copying DNA and help cells respond to stimuli. Using proteomics, it is possible to examine a large number of the different proteins in a tissue, which can provide information about the state of that tissue. Ubaida-Mohien et al. used this approach to answer the question of why muscles become weaker with age.

First, they analyzed the levels of all the proteins found in skeletal muscle collected from 58 healthy volunteers between 20 and 87 years of age. This revealed that the muscles of older people have fewer copies of the proteins that make up ribosomes – the cellular machines that produce new proteins – and fewer proteins involved in providing the cell with chemical energy. In contrast, proteins implicated in the immune system, in the maintenance of existing proteins, and in processing other molecules called RNAs were more abundant in older muscles.

Ubaida-Mohien et al. then looked more closely at changes involving RNA processing. Cells make proteins by copying DNA sequences into an RNA template and using this template to instruct the ribosomes on how to make the specific protein. Before the RNA can be 'read' by a ribosome, however, some parts must be cut out and others added, which can lead to different versions of the final RNA, also known as alternative transcripts.

In order to check whether the difference in the levels of proteins that process RNAs was affecting the RNAs being produced, Ubaida-Mohien et al. extracted the RNAs from older and younger muscles and compared them. This showed that the RNA in older people had more alternative transcripts, confirming that the change in protein levels was having downstream effects.

Currently, it is not possible to prevent or delay the loss of muscle strength associated with aging. Understanding how the protein make-up of muscles changes as humans grow older may help find new ways to prevent and perhaps even reverse this decline.
DOI: https://doi.org/10.7554/eLife.49874.002

capacity with aging is essential to identify new targets of intervention. Previous studies attempted to address this question by performing cross-sectional untargeted proteomic analysis in skeletal muscle biopsy specimens from young and old individuals. However, these studies were limited in size, focused on cancer cachexia, analyzed single fibers, did not account for levels of physical activity or did not explore the effect of aging over its continuous range and, therefore, could not distinguish changes due to aging from those due to disease or sedentary state (*Doran et al., 2009*; *Murgia et al., 2017*; *Waldera-Lupa et al., 2014*; *Brocca et al., 2017*; *Ebhardt et al., 2017*). To overcome these earlier limitations, we have performed a quantitative mass spectrometry-based proteome analysis (tandem mass tag, TMT) of skeletal muscle biopsies obtained from individuals distributed over a wide age range, who were healthy based on strict objective clinical criteria. We characterized proteins that were overrepresented and underrepresented in older individuals and using these data we made inferences about molecular pathways affected by aging in skeletal muscle.

## Results and discussion

### Quantitative skeletal muscle proteome analysis of healthy aging

Skeletal muscle biopsies were collected from 60 healthy participants of the Genetic and Epigenetic Study of Aging Laboratory Testing (GESTALT) aged 20 to 87 years who were defined as 'healthy' based on very strict evaluation criteria at the National Institute on Aging Clinical Unit in Baltimore (*Tanaka et al., 2018*). Exclusion criteria included any diseases that required chronic treatment (except for mild hypertension fully controlled with one drug only), any physical or cognitive

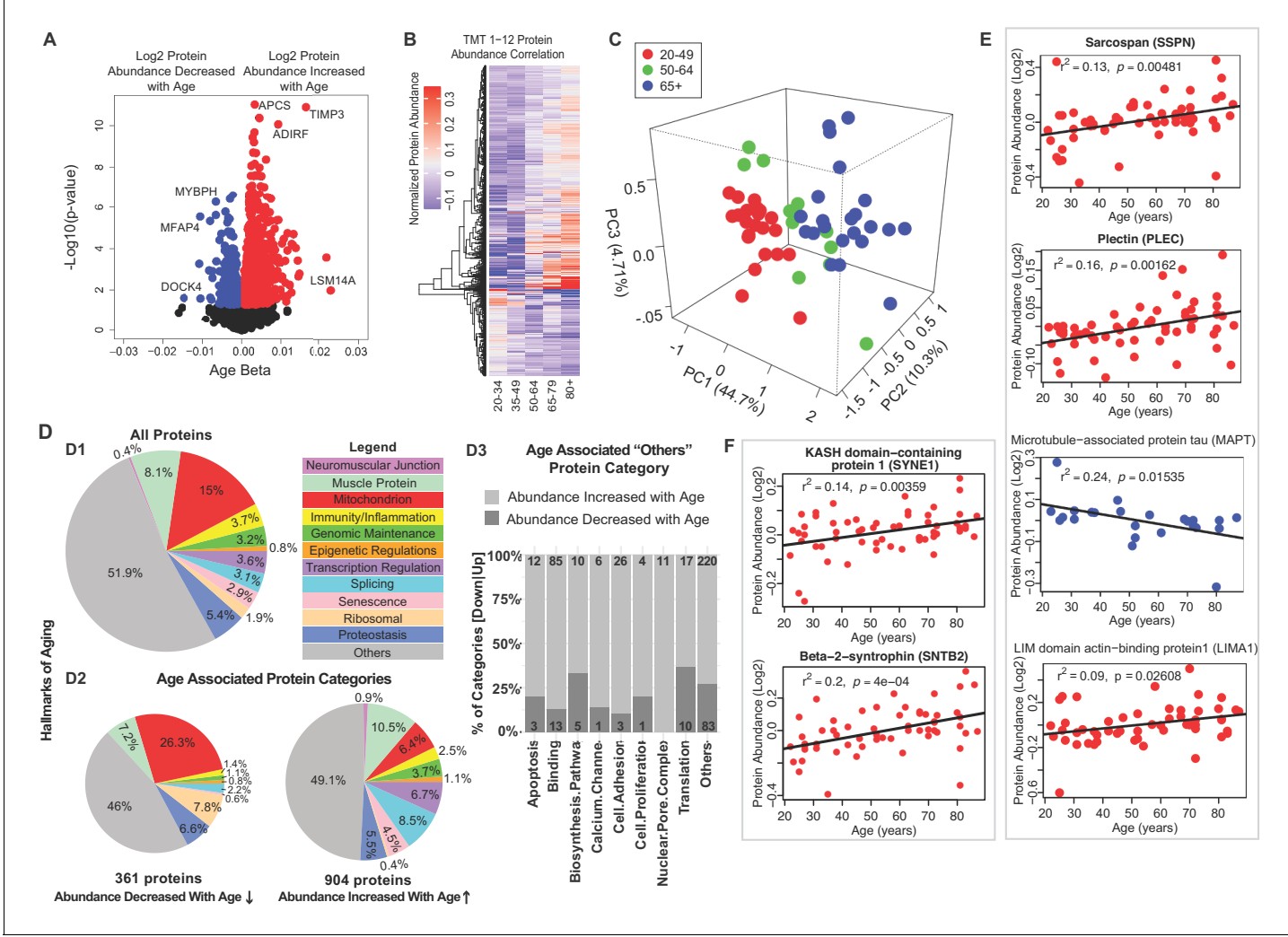

**Figure 1.** Classification of age-associated proteins. (**A**) Effect of age on protein expression levels. The x-axis represents the size and sign of the beta coefficient of the specific protein regressed to age (adjusted for covariates) and the y-axis represents the relative -log10 p-value. Each dot is a protein and all significant proteins are indicated in blue and red (age-associated 1265 proteins, p<0.05). (**B**) The heatmap of the 1265 significantly age-associated proteins reveals changing expression profiles across aging. (**C**) PLS analysis of age-associated proteins were classified into three age groups: 20–49 (young), 50–64 years (middle age), and 65+ (old) years old. (**D**) Percent distribution of categories of all quantified proteins, percent distribution of the same categories among proteins that were significantly downregulated and upregulated with aging. Proteins which are not considered directly related to mechanisms of aging are annotated as others and their subclassification is shown in the bar plot. (**E–F**) Log2 protein abundance of contractile, architectural and NMJ proteins. Simple linear regression was shown for age (x-axis) and protein (y-axis) correlation, confounders were not adjusted, and raw p-values were shown.

DOI: https://doi.org/10.7554/eLife.49874.003

The following source data and figure supplements are available for figure 1:

**Source data 1.** Baseline characteristics of the GESTALT skeletal muscle participants.
DOI: https://doi.org/10.7554/eLife.49874.009
**Source data 2.** Characteristics of participants.
DOI: https://doi.org/10.7554/eLife.49874.010
**Source data 3.** Complete protein dataset of skeletal muscle proteome quantified by TMT6plex.
DOI: https://doi.org/10.7554/eLife.49874.011
**Source data 4.** Complete peptide dataset of skeletal muscle proteome quantified by TMT6-plex.
DOI: https://doi.org/10.7554/eLife.49874.012
**Source data 5.** Dysregulated proteins with age.
DOI: https://doi.org/10.7554/eLife.49874.013

**Figure supplement 1.** Quantitative analysis of the skeletal muscle proteome with healthy aging.

*Figure 1 continued on next page*

*Figure 1 continued*

DOI: https://doi.org/10.7554/eLife.49874.004

**Figure supplement 2.** Quantitative analysis of muscle proteome.

DOI: https://doi.org/10.7554/eLife.49874.005

**Figure supplement 3.** Muscle proteins and robustness of age association.

DOI: https://doi.org/10.7554/eLife.49874.006

**Figure supplement 4.** Disregulation of proteins involved in genomic maintenance and cellular senesncence.

DOI: https://doi.org/10.7554/eLife.49874.007

**Figure supplement 5.** Age-associated ribosomal biogenesis proteins.

DOI: https://doi.org/10.7554/eLife.49874.008

impairment, and any abnormal values in pre-defined list of blood clinical tests (see Materials and methods for details). Participants who consented for a muscle biopsy were homogeneously distributed across the age strata 20–34 (n = 13), 35–49 (n = 11), 50–64 (n = 12), 65–79 (n = 12) and 80+ (n = 10), and biopsies were analyzed by tandem shotgun mass spectrometry-based quantitative proteomics method (*Figure 1—figure supplement 1*, *Figure 1—figure supplement 1*, *Figure 1—source data 2*). Using multiplexed isobaric labeling tags (TMT) and a customized analytical strategy (*Kammers et al., 2015*; *Herbrich et al., 2013*), we identified 400,000 tryptic peptides from 6.7 million spectra (396 multiplexed MS runs from 12 TMT 6-plex experiments), which allowed the quantification of 5891 proteins (*Figure 1—source data 1*, PXD011967).

To control for batch variability and avoid bias, we included a reference sample in all 12 TMT sets. A loading normalization was implemented that assumed that the sums of all intensities from all the proteins across the samples in a single TMT experiment were equal and that the sample loading effects, peptide bias effects and the residual error were normally distributed across a constant variance across samples (*Figure 1—figure supplement 2A*). To test the effectiveness of these approaches, we examined TMT batch effects in several analyses, allowing for experiment-specific random effects. We then averaged the expression values from each TMT across the sample groups and found that the ranks between TMTs were highly correlated (*Figure 1—figure supplement 1B–C*, *Figure 1—figure supplement 2B*). Together, these findings indicate that the protein quantification across the 12 TMT experiments was robust.

Of the initial 5891 proteins detected, we excluded from the analysis 1511 proteins that were not quantifiable in at least three participants per age strata (at least 15 participants total) and performed the analysis in the remaining 4380 proteins detected in more than 15 donors (three per age strata), which were quantified from 46,834 unique peptides and 2.7 million spectra (*Figure 1—figure supplement 1D*). We used Partial Least Squares (PLS) analysis to explore the overall clustering of the 4380 proteins across age groups (*Figure 1—figure supplement 1E*). The age groups (color-coded) were well separated along at least one axis in the three-dimensional clustering classification (*Figure 1—figure supplement 1C*). As expected, most of the proteins identified were classified as 'muscle proteins', and the top 10 most abundant muscle proteins accounted for 45% of the total spectral abundance (*Figure 1—figure supplement 2C*). Low-abundant mitochondrial proteins, such as cardiac phospholamban (PLN), were also quantified.

## Focus on the aging biological mechanisms

The relationship of age with the skeletal muscle proteins was estimated by linear mixed regression models that included sex, race, level of physical activity, type I/type II muscle fiber ratio, body mass index (BMI) and TMT batch effect as covariates (Materials and method). Of note, the age beta-coefficients (aging effect size) are small because they express the difference in protein 'per year' of age. For example, the difference in protein between two individuals that differ by 20 years would be 20 times the size of the beta coefficient. We adjusted for physical activity because it both tends to decline with age and strongly affects biological processes in muscle cells (*Egan and Zierath, 2013*; *McGee and Walder, 2017*; *Bauman et al., 2016*). Previous studies demonstrated that gender and race strongly affect body composition and muscle mass (*Gallagher et al., 1997*). Skeletal muscle tissue includes different myofiber types: type I fibers (slow-twitch), type IIa fibers (fast-oxidative), and IIb fibers (fast glycolytic muscle fibers containing four different myosin isoforms), each supported by different energetic metabolism and with different protein composition. An analysis for a proxy

measure of 'muscle fiber ratio' (*Schiaffino and Reggiani, 2011*) was estimated by calculating the ratio of myosin 7 (MYH7), the slow-twitch fiber isoform, and the sum of fast-twitch fiber isoforms (MYH1, MYH2 and MYH4) (*Figure 1—figure supplement 3 A1-A4*); as expected, the fiber ratio of slow/fast was higher with older age (*Figure 1—figure supplement 3 A5*). The slight change of slow/fast fiber ratio was significant and outweighed the wide variability among individuals (p=0.005); BMI was adjusted because obese persons tend to have muscle fat infiltration and lower muscle quality and muscle-fat interaction may affect muscle composition and function (*Moore et al., 2014*; *Silva and Martins, 2017*). Gender may also have an impact on protein expression in skeletal muscle, as males and females are known to have differences in muscle mass; however, because of the limited sample size, we did not stratify the analysis by gender. This analysis should be done in future larger studies.

Proteins were then deemed significantly underrepresented or overrepresented in older age based on *p*-values for age-coefficients in the regression equation, calculated from Satterthwaite's t-tests (*Figure 1—figure supplement 3B*). There were 1265 proteins significantly associated with age (p<0.05, with BH correction <0.1, 917 proteins), suggesting that approximately 29% of the skeletal muscle proteome changed with aging after 20 years of age (*Figure 1—source data 3*). Of these, 29% (361) were significantly underrepresented and 71% (904) were overrepresented with older age. The age-associated analysis across the experimental dataset and across multiple comparisons was highly robust (*Figure 1—figure supplement 3C*).

Notably, the proteins most strongly associated with older age (*Figure 1A*, right) were LSM14 homolog A (LSm14A, β = 0.023, p=0.0109), tissue metalloproteinase inhibitor 3 (TIMP-3, β = 0.0219, p=0.00026) and serum amyloid P-component (APCS, β = 0.0164, p=1.26E-11). Protein LSm14A is implicated in processing the assembly of processing bodies, involved in mRNA turnover, and can also bind to viral nucleic acids and initiate IFN-β production, contributing to innate immunity (*Li et al., 2012*). TIMP3 regulates the adipogenic differentiation of fibro/adipogenic progenitors (FAPs) in skeletal muscle, and its overrepresentation may explain the tendency for fat infiltration in aging muscle (*Kopinke et al., 2017*). Consistent with this hypothesis, Perilipin 1 (PLIN1, β = 0.014, p=0.0003), a lipid droplet-coating protein, and adipogenesis regulatory factor (ADIRF, β = 0.01173, p=1.78E-05), a protein that is only expressed in adipose tissue, were among the most overrepresented proteins in old muscle. APCS is indicative of systemic amyloid, and its overrepresentation in aging muscle has never been previously described.

The most underrepresented proteins (age β <−0.01 and p<0.05) with old age (*Figure 1A*, left) were HLA class II histocompatibility antigen (HLA-DRB1, p=0.024), dedicator of cytokinesis protein 4 (DOCK4, p=0.025), myosin-binding protein H (MYBPH, p=0.0005), and microfibril-associated glycoprotein 4 (MFAP4, p=0.000002). Although HLA-DRB1 is the most altered protein, it is present only in 53% of the donors. MYBPH maintains the structural integrity of the muscle and its decreased expression has been associated with muscle weakness in age-related disorders (*Hundley et al., 2006*).

To explore differences of protein expression profiles across the lifespan, we generated a heatmap of the 1265 age-associated proteins and looked for clusters of proteins showing parallel changes with age (*Figure 1B*). Hierarchical clustering of protein expression suggested that the strongest difference was between young (20-34) and old (80+). There were small differences before the age of 50, but afterwards there was on average three log fold protein expression differences, and even more substantial differences after the age of 64. The separation of protein expression among three age groups (20–49, 50–64, and 65+) was confirmed by PLS analysis (*Figure 1C*). These findings are consistent with changes in energy metabolism observed in rhesus monkeys and attributed to changes in PGC-1α-driven mitochondrial biogenesis, and with data showing that the age-associated decline of muscle strength is already detectable during the fourth decade of life and substantially accelerates after the age of 70 (*Dodds et al., 2014*; *Pugh et al., 2013*).

Next, we grouped all quantified proteins according to the main biological mechanisms of aging (*Figure 1*D.1). Because the tissue examined was skeletal muscle, we also included a category for all contractile and architectural muscle proteins (named hereafter 'muscle proteins'). Though the highest abundance proteins detected correspond to muscle proteins, the largest category were mitochondria proteins (15%). Each of the other categories represented <9% of total proteins. Protein classes that differed between those underrepresented and overrepresented with older age are summarized (*Figure 1*D.2 and 1D.3) and are described in detail in subsequent sections. Specifically,

proteins implicated in muscle contraction, muscle architecture, mitochondria metabolism, as well as ribosome function decreased with older age. By contrast, proteins related to genomic maintenance, transcriptional regulators, splicing, neuromuscular junction, proteostasis, senescence and immune function increased with age. Other smaller subcategories of proteins were also differentially abundant in older muscle (*Figure 1*D.3).

## Contractile, architectural and neuromuscular junction proteins (NMJ)

Since many proteins decreasing with age were contractile proteins, we classified these further by function. The top 95 proteins in this class are involved in the architectural and functional stabilization of the sarcomere, including sarcospan (SSPN, β = 0.002, p=0.016) (*Figure 1E*), a dystrophin-associated protein complex important for muscle regeneration, actin-binding LIM domain and actin-binding protein 1 (LIMA1, β = 0.003, p=0.009), a cytoskeleton-binding protein that stabilizes actin filaments, and plectin (PLEC, β = 0.0007, p=0.036), a large cytoskeleton protein that preserves interactions within the acto-myosin complex. Increases in delta sarcoglycan (SGCD, β = 0.0019, p=0.00004), and gamma sarcoglycan (SGCG, β = 0.0016, p=0.0062) were consistent with mouse studies showing that dystrophin, sarcoglycan subcomplex γ- and δ-sarcoglycan were overexpressed with aging, perhaps a compensatory mechanism to avoid damage in the sarcomere during contraction or as biomarkers of continuous repair (*Hughes et al., 2015*). Interestingly, MAPT (tau, mostly expressed in neurons and involved in the assembly and stabilization of microtubules), was also significantly underrepresented in older muscle (*Figure 1E*). A crucial component of muscle function is the neuromuscular junction (NMJ), and since the abundance of all NMJ-related proteins increased with age we examined the agrin signaling pathway of NMJ. Agrin (AGRN) and acetylcholine esterase (ACHE) increased with age but not significantly (*Figure 1—figure supplement 3D*). By contrast, the levels of Syne-1 which anchors both synaptic and non-synaptic myonuclei for proper neuron innervation and respiration increased with age (SYNE1 β = 0.002, p=0.005) as did beta-2-syntrophin, which is believed to be involved in acetylcholine receptor clustering (SNTB2, β = 0.0029, p=0.0003).

## Decline of mitochondrial proteins with age

Because of the striking difference in abundance of mitochondrial and energy metabolism proteins with age, we studied these proteins by protein annotations using Uniprot keywords, GO ontology terms and extensive manual curation based on the most recent literature. The coverage of mitochondrial proteins quantified by our analysis compared to those described in the literature ranged from 92% for TCA proteins to 52% for proteins located in outer mitochondrial membrane [possibly due to incomplete tissue disruption (*Morgenstern et al., 2017*; *Zhao et al., 2014*)] (*Figure 2A*). The coverage of the bioenergetics and mitochondrial proteome in our dataset is similar to that reported by other authors (*Murgia et al., 2017*; *Morgenstern et al., 2017*). Of the mitochondrial proteins identified, the abundance of 25% of them (173 proteins) changed with age, mostly (70%) declining with age. Notably, however, outer membrane proteins were more abundant (*Figure 2B*); for example, NADH-cytochrome b5 reductase 3 (CYB5R3), an NADH-dehydrogenase located in the outer membrane of ER and mitochondria, whose overexpression is known to mimic many effects of caloric restriction, was significantly overrepresented in older age (*Figure 2—figure supplement 1A*) (*Martin-Montalvo et al., 2016*; *Diaz-Ruiz et al., 2018*). The permanence of mitochondrial protein debris in aging muscle has been previously reported - attributed to defective autophagy, and through to cause activation of the inflammasome and a proinflammatory state (*Ferrucci and Fabbri, 2018*).

Of the enzymatic mitochondrial proteins, 99 were respiratory chain proteins (Complex I-V and assembly complex proteins), and most of them declined with aging (28 proteins p<0.05; *Figure 2C*). Surprisingly, succinate dehydrogenase complex assembly factor 2 (SDHAF2), required for covalent FAD insertion into complex II, the electron transport chain, and the TCA cycle, were significantly overrepresented with older age (*Figure 2C* inset). The reason for this exception is unclear and if replicated in other analysis requires further work.

We then analyzed proteins from complex I to V and found that 16 proteins were significantly lower at older age (*Figure 2C*, *Figure 2—figure supplement 1B*). Among 41 proteins involved in energy production, most were underrepresented at older ages. Of 22 proteins quantified for TCA cycle, only malate dehydrogenase (MDH1), isocitrate dehydrogenase (IDH1), fumarate hydratase (FH) and succinate–CoA ligase (SUCLG1) (*Figure 2—figure supplement 1C*) were significantly lower

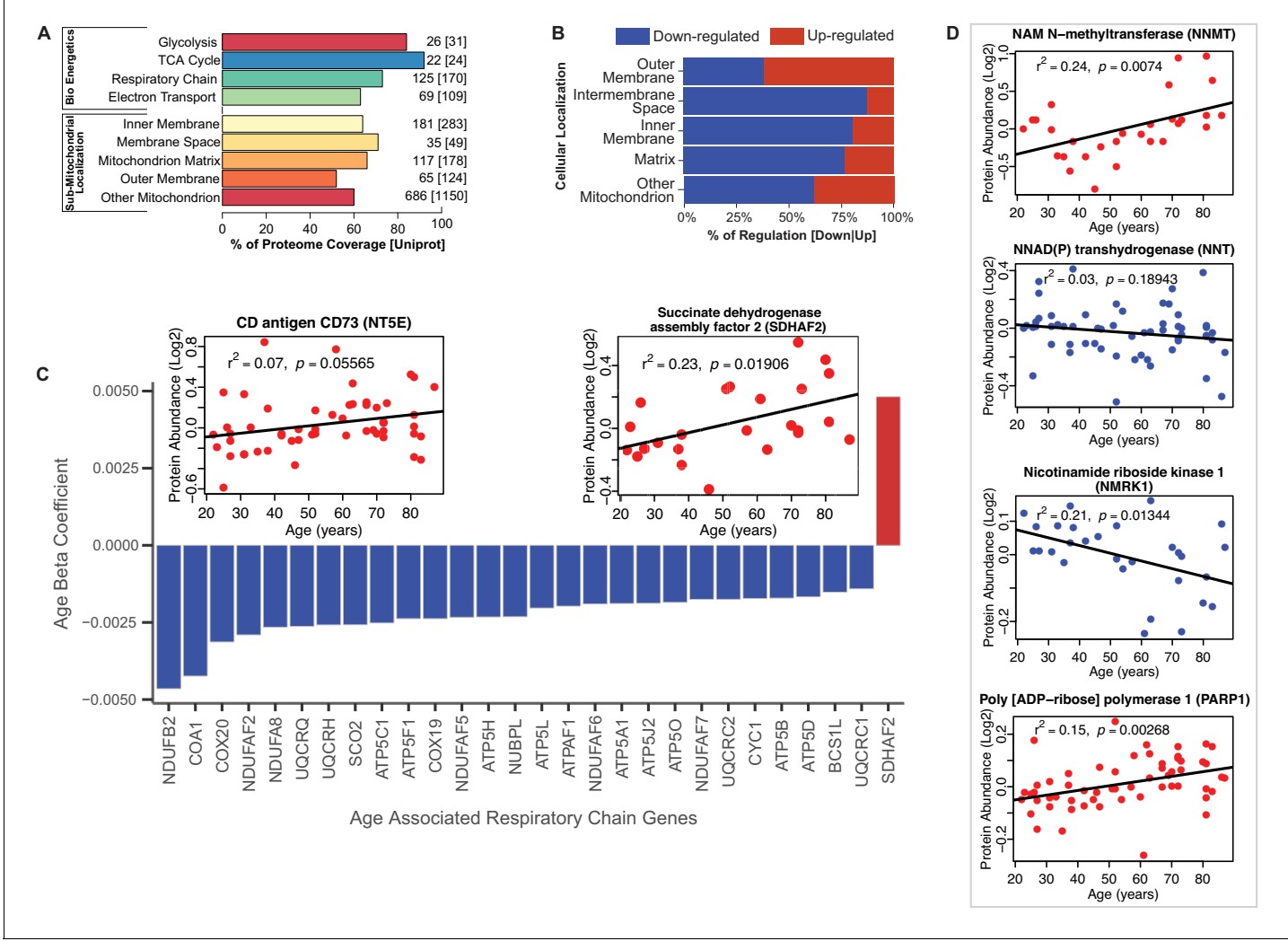

**Figure 2.** Functional decline of mitochondrial proteins with age. (A) Percent coverage within categories of skeletal muscle proteins compared to the Uniprot database. The top section shows various energetics categories, while the z axis indicates the number of proteins identified for each protein category and in parenthesis the number of proteins reported in Uniprot for the same category. (B) Subcellular location of age-associated mitochondrial proteins based on up- or downregulation. Of note, most of the mitochondrial proteins are downregulated. (C) Age-dependent decline of respiratory and electron transport chain proteins. All mitochondrial proteins in the respiratory and electron transport chain that are significantly associated with age are downregulated (p<0.05) except SDHAF2. The inset panel reports data on the proteins that are significantly upregulated with aging, SDHAF2 (mitochondrial) and the membrane protein CD73.

DOI: https://doi.org/10.7554/eLife.49874.014

The following figure supplement is available for figure 2:

**Figure supplement 1.** Age-associated bioenergetics pathways.

DOI: https://doi.org/10.7554/eLife.49874.015

at older ages. The decreasing levels of IDH-1 with age is unsurprising, as previous studies have shown a decrease in abundance of IDH-1 in older *C. elegans* (*Copes et al., 2015*). IDH1 converts isocitrate to α-ketoglutarate by reducing $NADP^+$ to NADPH in the process. In addition, to IDH1, $NADP^+$ is also reduced to NADPH via the mitochondrial NAD(P)-malic enzyme (ME2) (*Sauer et al., 2004*) and predominantly through NNT (NAD(P) transhydrogenase) and the pentose phosphate pathway. In our study, NNT (β = −0.003, p=0.001) significantly decreased with aging. Interestingly, the decrease in expression levels of both NNT and IDH1 with age, suggests a decreased capacity of the mitochondria to maintain proton gradients and results in oxidative damage. Further, NADK ($NAD^+$ Kinase), which is highly regulated by the redox state of the cell and regulates $NADP^+$

synthesis in vivo decreased with age (NADK2, β = −0.001, p=0.052). The changes in the $NADP^+$/NADPH levels influence cellular metabolism, calcium signaling and anti-inflammatory processes and regeneration of glutathione (*Sheeran et al., 2010*).

$NAD^+$ declines with age in several tissues and its metabolism has been implicated in the aging process and age-related pathologies including loss of skeletal muscle mass (*Fang et al., 2016*; *Goody and Henry, 2018*). $NAD^+$ is synthesized in vivo predominantly via the salvage pathway and the de novo and Preiss-Handler pathways (*Verdin, 2015*; *Bogan and Brenner, 2008*). We specifically examined age differences in abundance of proteins from these pathways. We found that NAM-N-methyl transferase (NNMT, β = 0.007, p=0.016), nicotinamide ribose kinases (NMRK1, β = −0.003, p=0.002), poly-ADP-ribose polymerases (PARP1, β = 0.002, p=0.003) and CD73 (NT5E, β = 0.004, p=0.056) were significantly increased with at older ages, while only NMRK1 decreased with age (*Figure 2D*). NAMPT, which converts NAM to NMN, was not significantly different with age while NMRK1, which converts NR to NMN, was significantly lower in the muscle of older participants. These findings may explain the mechanism by which NMN tends to be lower in tissue from older compared to younger persons. Two additional mechanisms may exacerbate the decline in NMN and $NAD^+$ with aging, namely the increased expression levels of CD73 that converts NMN to NR and the increase in expression levels of PARP1, which converts $NAD^+$ into NAM and ADP-ribose.

The findings described above are consistent with those reported in model organisms, including non-human primates, suggesting that changes in muscle with aging are characterized by profound changes in energy metabolism and, in particular, oxidative phosphorylation, because of either defects in mitochondrial biogenesis or impaired autophagy flux that hampers the recycling of dysfunctional mitochondria (*Pugh et al., 2013*; *Azzu and Valencak, 2017*). Changes in energy metabolism occur in parallel to changes in body composition, although it is still unclear what changes occur first (*Speakman and Westerterp, 2010*). In addition, changes similar to those mentioned above for mitochondria and intramuscular lipid metabolism were previously found in muscle from frail mice, suggesting that the pathogenesis of frailty in mice and humans may be interpreted as 'accelerated aging' (*Huang et al., 2019*). Also, we and others have previously demonstrated that changes in skeletal muscle proteins that occur with aging, including those identified in this study, are partially reversed by physical activity (*Ubaida-Mohien et al., 2019*; *Hood et al., 2019*). The partial overlap between the effects of aging and exercise suggests that despite opposite physiologic effects, aging and exercise affect skeletal muscle tissue though different root mechanisms (*Hood et al., 2019*). This conclusion is consistent with the finding reported by *Phillips et al. (2013)*, after analyzing gene expression patterns in humans.

## Changes of proteins involved in genomic maintenance and cellular senescence

Most genomic maintenance proteins increased in abundance with age, especially those involved in DNA damage recognition and repair, such as double-strand break repair protein (MRE11), X-ray repair cross-complementing protein 5 (XRCC5), and structural maintenance of chromosomes protein 1A (SMC1A) (*Figure 1—figure supplement 4A*). Prelamin-A/C (LMNA), Lamin-B1 (LMNB1) and Lamin-B2 (LMNB2), members of the LMN family of protein components of nuclear lamina that help maintain nuclear and genome architecture, were all overrepresented with older age (*Figure 1—figure supplement 4B*). Sirtuin 2 (SIRT2, β = −0.0013, p=0.032), implicated in genomic stability, metabolism and aging, was also found to be lower in older skeletal muscle (*Figure 1—figure supplement 4C*). The increased abundance of genomic maintenance proteins may represent an attempt to compensate for the accumulation of somatic mutations in myocytes, and especially in satellite cells, as previously demonstrated in humans (*Franco et al., 2018*).

Forty proteins that in the literature have been implicated in cellular senescence were significantly overrepresented with age. These included extracellular superoxide dismutase (SOD3, β = 0.005, p=0.000009) and Transgelin-2 (TAGLN2, β = 0.005, p=0.0002), a potential oncogenic factor and senescence-associated protein. Proteins that decreased with age were GOT1, MAP2K3 (β = −0.003 and −0.0021, respectively), and casein kinase II subunit alpha (CSNK2A1, β = −0.0016, p=0.014). Interestingly, in addition to regulating the cell cycle, CSNK2A1 plays a central role in many other biological mechanisms, including apoptosis, which is suppressed in senescent cells (*Figure 1—figure supplement 4D*). These observations suggest that senescent cells from different possible origins (e.g. myocytes, adipocytes or fibroblasts) may accumulate in old muscle. However, since previous

studies were unable to detect direct biomarkers of senescence in human skeletal muscle, we cannot exclude the possibility that proteins related to senescence overrepresented in this study were expressed in intramuscular fat or other non-muscular cells (*Justice et al., 2018*).

## Transcription and splicing

Of all the 69 age-associated transcription-regulatory proteins quantified, 61 were overrepresented and only eight were underrepresented with older age. This last subset included kelch-like protein 31 (KLHL31, β = −0.0017, p=0.003), which is essential for muscle development and perhaps also muscle repair (*Abou-Elhamd et al., 2009*). This protein is expressed before MYOD in developing skeletal muscle and contributes to myogenic commitment, probably by acting as a transcriptional regulator in the MAPK/JNK signaling pathway. Deficiency in KLHl31 causes congenital myopathy in mice (*Papizan et al., 2017*). Of note, the levels of myocyte-specific enhancer factor 2D (MEF2D, β = 0.003, p=0.018), essential for myogenesis and muscle regeneration and the primary regulator of KLHL31 production, increased, possibly as a compensatory mechanism (*Schiaffino et al., 2018*). Contrary to earlier reports, CTCF (β = 0.009, p=0.026), a transcriptional activator and repressor protein that fine-tunes chromatin architecture, also increased with age (*Figure 3A*).

A major unexpected finding of our analysis was the strong increase in the main spliceosome complex proteins with aging (*Figure 3—figure supplement 1A*). The spliceosome comprises five small nuclear RNAs (snRNAs), *U1*, *U2*, *U4*, *U5*, and *U6*, that form functional complexes with proteins to regulate alternative splicing, a process by which different exons of one pre-mRNA are variably combined to generate different proteins (*Papasaikas and Valcárcel, 2016*). We found differential expression of many proteins widely distributed across the five spliceosome complexes and other spliceosome-associated protein factors essential for mRNA maturation and gene expression (*Figure 3B*). In particular, of the ~300 proteins and spliceosome-associated factors described (*Zhou et al., 2002*; *Rappsilber et al., 2002*), we found that 99 and 57 of them, respectively, were overrepresented in older muscle (*Figure 3C*). Overall, spliceosomal proteins increased by ~15% between the ages of 20 and 87 years (*Figure 3D*). Spliceosome components are actively rearranged during assembly, catalysis, disassembly and recycling, each step involving recruitment and recycling of several proteins (*Wahl et al., 2009*).

To understand whether aging affects preferentially one of these biological steps, we categorized the spliceosomal complexes and snRNPs into E complex, A complex, and B complex (assembly complex, 37 proteins), Bact complex and C complex (catalysis complex, seven proteins) and snRNPs (recycling, SART1 protein) (*Figure 3E*), but we found no evidence of proteins from a specific complex being more overrepresented with aging than proteins from other complexes (*Figure 3—figure supplement 1B*). LSm RNA-binding protein (LSM14A) was the most overrepresented assembly protein, displaying a 20-fold increase with age. The overrepresentation of spliceosomal proteins, such as the pre-mRNA-processing-splicing factor 8 (PRPF8) (*Figure 3E*, inset), the key catalytic core and the largest and most conserved protein in the spliceosome, suggests that pre-RNA processing may be upregulated in older skeletal muscle.

Systematic changes in the splicing machinery with older age was previously suggested by epidemiological studies (*Holly et al., 2013*), transcriptomic analyses of skeletal muscle biopsies (*Giresi et al., 2005*; *Welle et al., 2003*), and human peripheral blood leukocytes (*Harries et al., 2011*) of young and old individuals. In these studies, processing of mRNAs was the feature that best discriminated between younger and older persons, suggesting that modulation of alternative splicing is one of the signatures of aging (*Latorre and Harries, 2017*). Although the mechanisms and consequences of the rise in splicing factors with aging are unknown, they may indicate either a dysregulation of the splicing apparatus or a shift toward increased splicing and/or altered splice isoform diversity with aging (*Welle et al., 2003*). Consistent with this view, a defect in alternative splicing is implicated in some fundamental mechanisms of aging, such as cellular senescence (*Latorre and Harries, 2017*; *Deschênes and Chabot, 2017*), as well as in many chronic, age-related conditions, such as Alzheimer Disease and cancer-related cachexia (*Raj et al., 2018*; *Narasimhan et al., 2018*).

## Age-associated alternative splicing and splicing events

The marked rise in overrepresentation of splicing machinery proteins with aging prompted questions about its functional consequences. Emerging literature suggest that change in expression of splicing

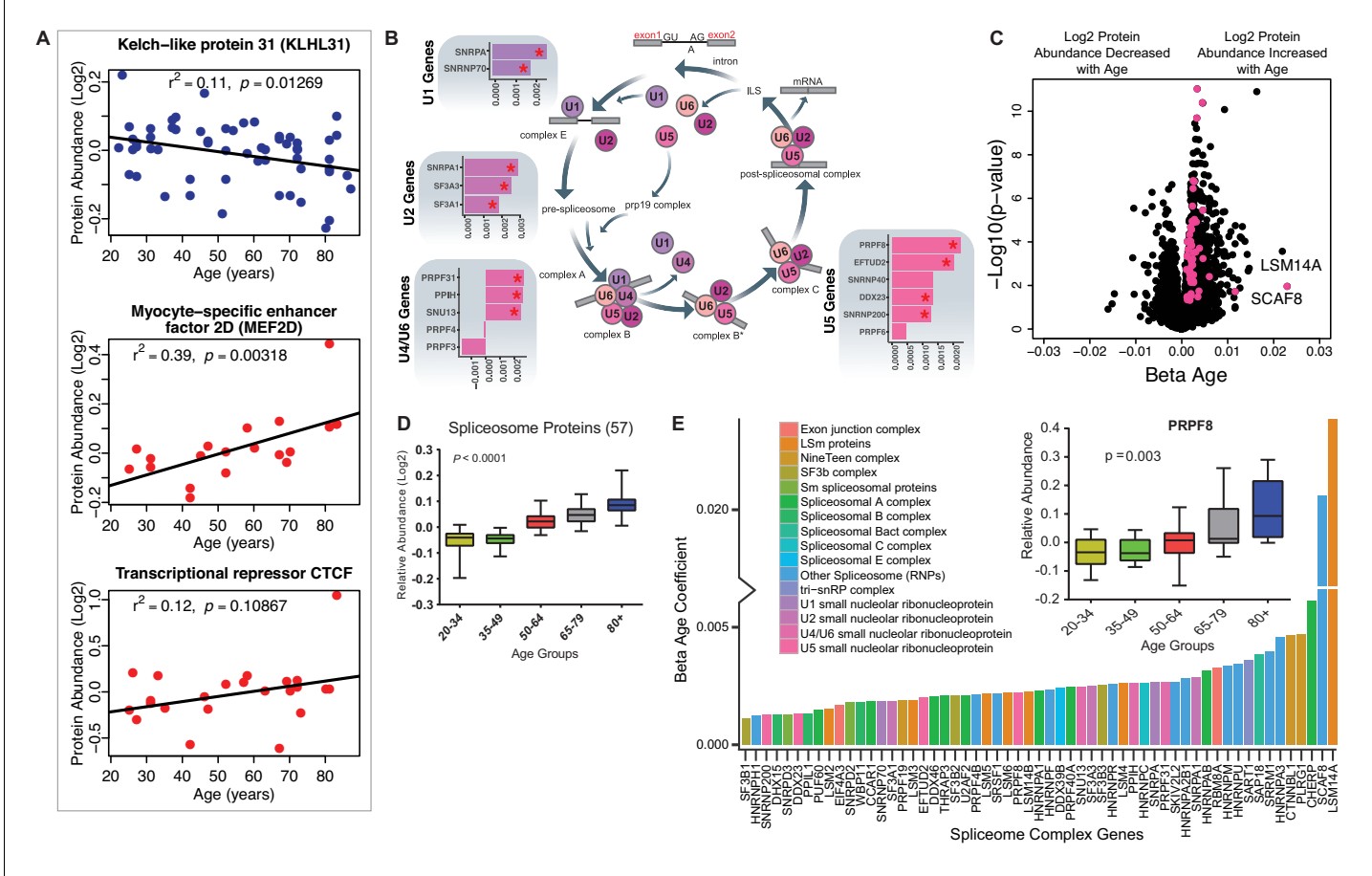

**Figure 3.** Implications of proteins that modulate transcription and splicing. (A) Log2 protein abundance of age-associated transcriptional regulation proteins. Simple linear regression was shown for age (x axis) and protein (y axis) correlation, unadjusted p-values were shown. (B) Spliceosome major complex pathway protein expression abundance and dysregulation. KEGG major spliceosome complex pathway representation and spliceosome complex proteins quantified (associated with splicing RNAs *U1*, *U2*, *U4/U6*, and *U5*) as plotted in the side square boxes. (C) The log2 abundance expression of 57 spliceosome complex proteins associated with age (p<0.05) are depicted as magenta circles, while all other quantified proteins are black circles. All snRNPs and spliceosome regulatory proteins are upregulated with age. (D) The average of all age-associated spliceosome proteins within each age group reveals an upregulation of spliceosome proteins with age. (E) Effect of age (one-year difference) on the 57 proteins of the spliceosome major complex and color coded based on spliceosome domains. Inset (left) is a legend for the complex domains and inset (right) shows that PRPF8 protein is robustly overrepresented with age.

DOI: https://doi.org/10.7554/eLife.49874.016

The following figure supplement is available for figure 3:

**Figure supplement 1.** Spliceosomal proteins and age association.

DOI: https://doi.org/10.7554/eLife.49874.017

factors is a major determinant for selection of specific splicing variants and changes in splicing variants contributes to some aging phenotypes, including age-related diseases (*Deschênes and Chabot, 2017*; *Mazin et al., 2013*). We analyzed potential differences in mRNA splicing with age (see Materials and methods) using RNA-seq data that were available for most of the same specimens used for the proteomic study (n = 53). Specifically, we studied a set of variations of the exon-intron structure, known as transcriptional events, namely Alternative First exon (AF), Skipped Exon (SE), Alternative Last exon (AL), Alternative 3' splice-site (A3), Alternative 5' splice-site (A5), Retained Intron (RI) and Mutually Exclusive Exons (MX) (*Alamancos et al., 2015*). Donor-specific splicing index (PSI, which measures each isoform as a % of total isoforms) was calculated for each AS event in each sample and a linear mixed regression model was used to identify age-associated PSIs for each splicing event. Analysis of 144,830 transcripts from RNA-seq datasets showed that around 3.7% of the skeletal muscle transcripts (5459 transcripts, corresponding to 6255 events) showed relative

abundance changes with aging (*Figure 4A*; *Figure 4—source data 1*). Next, we calculated the frequency and distribution of splicing events with aging as well as the directionality of such changes and found that 2714 events were significantly less frequent at older ages and 3545 events significantly more frequent at older ages (*Figure 4B*; *Figure 4—source data 2*). The overall number of

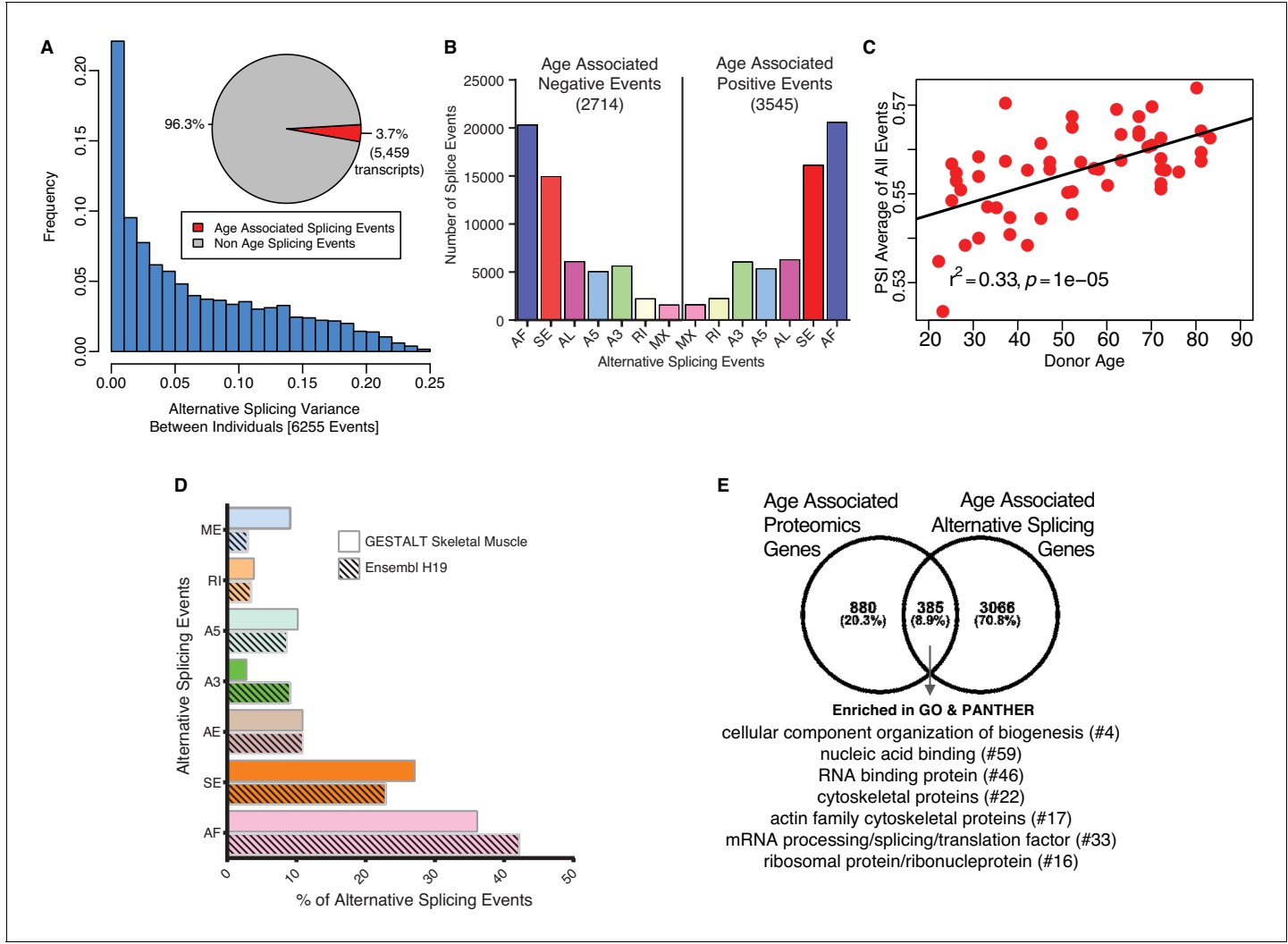

**Figure 4.** Age-associated alternative splicing. (A) The number of participants with detected splicing variants is substantial, with >20% of the participants showing <5% splicing variants for the detected gene. Overall, 3.7% of all identified skeletal muscle transcripts (3791 genes) show evidence of differential alternative splicing with aging. (B) Comparison of skeletal muscle age-associated splicing events (positive and negative). Negative events are downregulated with age and positive events are upregulated with age. The category of event is plotted on the x-axis, and the y-axis shows the number of splicing events for each category. (C) Average Percent Spliced-in (PSI) from 6255 events for each donor is depicted as a red circle. Average PSI across 53 donors ranging in age from 20 to 84 suggests an increase in alternative splicing with aging. (D) Comparison between skeletal muscle splicing events detected in GESTALT (solid bars) and splicing events reported in the Ensemble transcript splice events (shaded bars). (E) Comparison of age-associated proteins and age-associated alternatively splicing genes suggesting 30% (385) of the age-associated proteins undergo alternative splicing. Analysis of PANTHER database shows classes of enriched genes from different protein categories, and the number of genes representing each class is shown with #.

DOI: https://doi.org/10.7554/eLife.49874.018

The following source data is available for figure 4:

**Source data 1.** Age-associated splicing events.
DOI: https://doi.org/10.7554/eLife.49874.019

**Source data 2.** Age-associated positive and negative splicing events.
DOI: https://doi.org/10.7554/eLife.49874.020

events increased slightly with older age but AS events, at least for the 6,255 AS events quantified, increased significantly with age (r2 = 0.33, p=6.001e-06) (*Figure 4C*).

We then investigated whether any specific class of skeletal muscle AS events was enriched in our age-association analysis compared to the list of splicing events described for human skeletal muscle in the Ensembl human transcriptome (*Figure 4D*). The rates of observed skeletal muscle events are very similar to those reported in the Ensembl transcriptome (*Figure 4D*) except for ME, A3, SE and AF. The largest difference was in the skipped exon (SE) class of events, where a higher percentage of transcripts were exon-skipped compared to Ensembl events, with 27% of all the skeletal muscle AS events of the exon skipping type. A previous study reported 35% of the erythroid genes show evidence of AF exons, indicating that alternative promoters and AF are widespread in the human genome and play a major role in regulating expression of select isoforms in a tissue-specific manner (*Tan et al., 2006*). This finding is in line with our result of 36% AF in our skeletal muscle data.

We next examined whether AS events occurred in proteins connected with pathways that are known to be dysregulated with aging; interestingly, among the top fold enriched (FE) gene ontology (GO) biological processes associated with age, splicing changes were more frequent on those that negatively regulated IκB kinase/NF-κB signaling (FE = 2.86, p=3.18E-04), and those that regulated mitophagy (autophagy of mitochondria; FE = 3.71, p=2.23E-04) and fatty acid beta oxidation (FE = 3.21, p=1.72E-04). The GO biological process with positive age-associated splicing events were mitochondrial morphogenesis (FE = 5.15, p=8.98E-03), response to mitochondrial depolarization (FE = 4.93, p=2.46E-04), and endoplasmic reticulum calcium ion homeostasis (FE = 4.48, p=2.31E-04). These data suggest that the upregulation of alternative splicing in skeletal muscle with aging may react to change that occur with aging either by rising an inflammatory response or by activating damage-response mechanisms at a time when energy becomes scarce.

Among the 5459 transcripts (from 3791 genes) that were alternatively spliced with age, 4967 transcripts were protein-coding. We compared these genes with the age-associated proteins and found that 8.9% of the age-associated alternatively spliced transcripts (385) were reflected in protein changes (*Figure 4E*). This comparison of age-associated proteins and alternatively splicing mRNAs suggests that 30% (385) of the age-associated proteins undergo alternative splicing. Among this group, 64 proteins are involved in cellular organization or biogenesis (GO:007180), and proteins like tubulin (TUBB2B, TUBB), profilin 2 (PFN2) and actin-related protein 2/3 complex subunit 4 (ARPC4) are involved in the cytoskeletal regulation by Rho GTPase pathway. A further PANTHER database classification of these proteins shows an enrichment in categories like RNA/DNA binding, cytoskeletal, translational and ribosomal proteins (*Figure 4E* protein categories). Overall, these findings suggest that a large percentage of proteins that change with aging in muscle also undergo splicing variations, and this is especially true for mitochondrial proteins, perhaps as a resilient response to the energetic deficit that develops with aging. This hypothesis is consistent with several lines of research suggesting that mechanisms of alternative splicing are enhanced in tissues that are highly energetically demanding, such as muscle and brain (*Pan et al., 2008*). Also, higher physical activity has been associated with downregulation of proteins from the splicing machinery (*Ubaida-Mohien et al., 2019*).

## Depletion of ribosomal proteins with age

Similarly, to previous studies, we found that a large number of ribosomal proteins were differentially expressed with older age (*Figure 1*, *Figure 1—figure supplement 5A*) (*Steffen and Dillin, 2016*; *Kirby et al., 2015*). In particular, all the 60S and 40S ribosomal proteins were globally reduced in older muscle; exceptions included 60S ribosomal proteins RPL12 and RPL3 (RPL12, β = 0.0008, p=0.024, RPL3, β = 0.003, p=0.016), as well as H/ACA ribonucleoprotein complex subunit 4 (DKC1, β = 0.002, p=0.034) and nucleolar protein 58 (NOP58, β = 0.003, p=0.00007), which were overrepresented in old muscle. RPL12, RPL3, and DKC1 play a role in viral mRNA translation, while NOP58 is important for ribosomal biogenesis (*Figure 1—figure supplement 5A-C*). Changes in ribosome proteins may signal lower ribosomal biogenesis with aging in skeletal muscle, with ensuing decline in protein synthesis with aging (*Turi et al., 2019*). Over time, this defect may lead to slow turnover and progressive damage accumulation in contractile proteins.

# Differential regulation of proteins related to proteostasis in aging

Cells rely on a complex proteostatic machinery to handle protein quality control, assembly, folding and elimination. These activities are essential for the recycling of damaged proteins or entire organelles and provide critical protection against damage during conditions of metabolic or oxidative stress. Extensive literature supports the decline of proteostasis with aging in animal models and in humans (*Kaushik and Cuervo, 2015*; *Charmpilas et al., 2017*). Of the 239 detected proteins that has been related to proteostasis in the literature, 31% were altered with age (p<0.05, 24

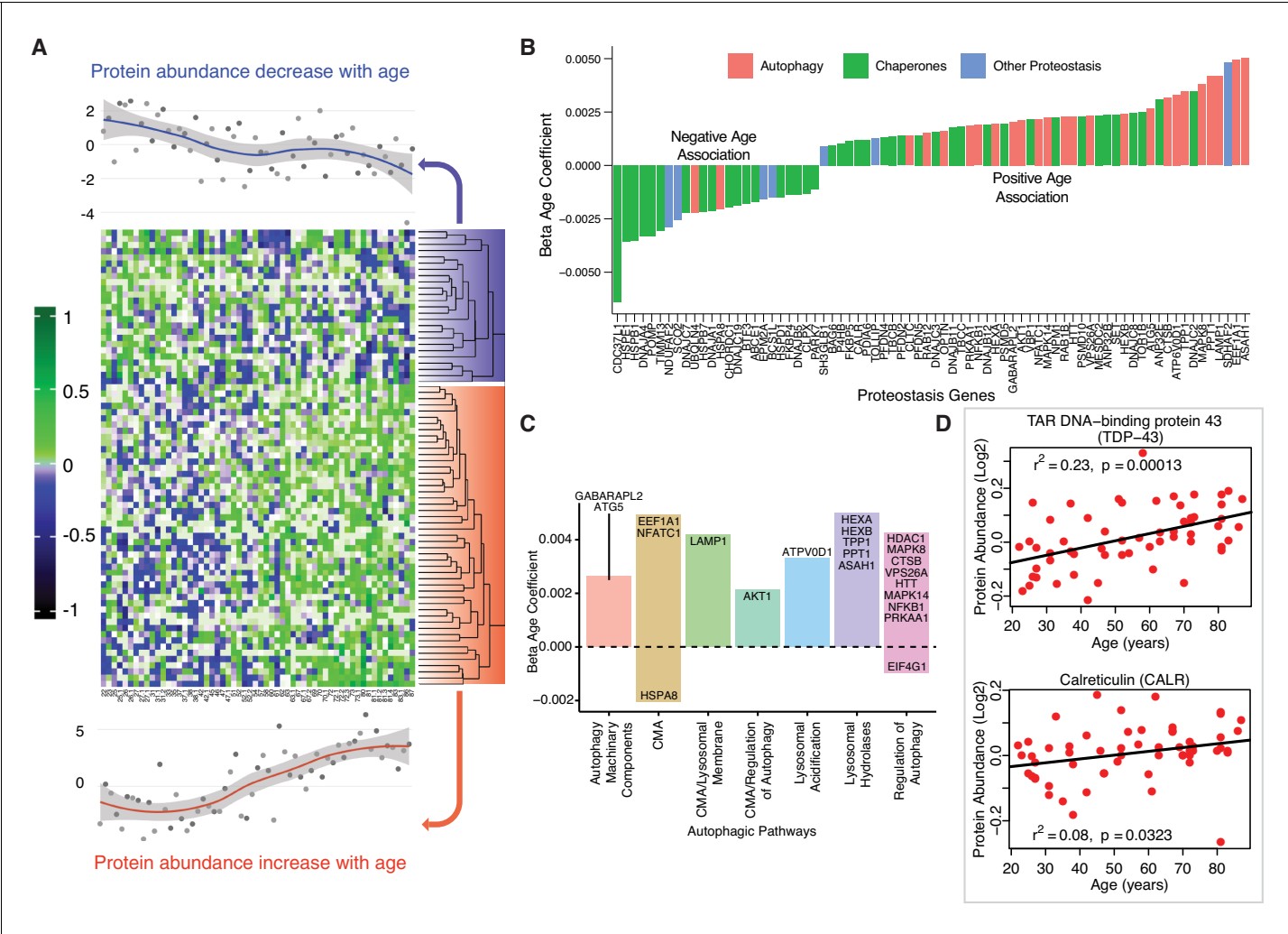

**Figure 5.** Age-associated proteostasis proteins. (A) Log2 protein abundance of all 74 age-associated proteostasis proteins across all 58 donors. Rows represent proteostasis proteins and columns represents donors. The average expression of all age-associated proteostasis proteins from each donor is plotted at the top and bottom (gray circles) with loess curves. The top section shows downregulated clusters of proteins (average of 24 proteins) and the bottom shows upregulated clusters of proteins (average of 54 proteins). The protein rows are ordered based on hierarchical clustering and displayed by dendrograms. (B) Confounders-adjusted β age coefficient of age-associated proteins, showing age-differentially regulated proteostasis proteins, over representation of proteostasis category proteins and the log2 magnitude of protein change with each year of age. (C) The increase of autophagy protein sub category with age is shown. Except HSPA8 and EIF4G1 all other autophagy proteins are positively correlated with age. Each bar plot shows each autophagy protein sub category and the average change over a year of age. The gene/proteins names are organized from lowest to highest log2 expression change per year of age. (D) Raw log2 abundance of autophagy proteins TDP and CALR were shown, simple linear regression method was used for age (x axis) and protein (y axis) correlation, of note unadjusted p-values were shown.

DOI: https://doi.org/10.7554/eLife.49874.021

The following figure supplement is available for figure 5:

**Figure supplement 1.** HSPA8 protein and its association with age.

DOI: https://doi.org/10.7554/eLife.49874.022

underrepresented and 50 overrepresented with older age) (*Figure 5A*). Most proteins underrepresented with age were chaperones, including DnaJ homolog subfamily A (DNAJA1), also named heat shock protein (Hsp) 40 (β = −0.0021, p=0.003), Hsp27 (β = −0.004, p=0.0001), Hsp70 protein 8 (HSPA8, β = −0.002, p=2.34E-07) (*Figure 5—figure supplement 1*) as well as Hsp27 protein 1 (HSPB1) and protein 7 (HSPB7), Hsp10 protein 1 (HSPE1) and Hsp60 protein 1 (HSPD1). Excluding HSPD1, the decline of these proteins with aging was previously described (*Charmpilas et al., 2017*; *Senf, 2013*; *Matsumoto et al., 2015*). Other differentially expressed proteostasis-related proteins, including PDIA6, NPM1, ANP32E, and DNAJC2, are also regulatory chaperones (*Figure 5B*).

The loss of chaperone function during aging may be compensated by an increase in autophagic activity (*Figure 5C*), as misfolded proteins must be removed and degraded through an alternative mechanism. Indeed, most proteostasis proteins overrepresented with aging were related to autophagy except HSPA8 and Eukaryotic translation initiation factor 4 gamma 1 (EIF4G1). For example, TDP-43, a DNA/RNA-binding protein that tends to form aggregates in tissues such as skeletal muscle and brain and is both removed by autophagy and involved in autophagy maintenance, increased significantly with aging (TARDBP, β = 0.002, p=0.0002) (*Figure 5D*). On the contrary, calreticulin (CALR), a quality control chaperone induced under ER stress that stimulated autophagy, was significantly higher in muscle of older participants (β = 0.001, p=0.022) (*Figure 5D*) (*Yang et al., 2019*). Of note, calreticulin is used by macrophages to tag cells to be removed by programmed cell phagocytosis (*Krysko et al., 2018*). Consistent with this finding, CALR is considered a main biomarker of age-related diseases and frailty (*Cardoso et al., 2018*). Overall, our findings are consistent with the ample evidence in the literature that aging is associated with a decline in chaperone-mediated autophagy, and that proteostatic mechanisms are important for aging and longevity (*Cuervo and Wong, 2014*; *Sands et al., 2017*). However, we also found in older muscle a general increase in proteins implicated in macroautophagy, possibly representing a compensatory mechanism.

## Pro-inflammatory and anti-inflammatory immune proteins of aging muscle

Of the 32 immune-related age-associated proteins that were quantified (*Figure 6A*), three broad themes emerged from the aging muscle immune proteome (*Figure 6B–D*). First, many proteins previously linked to macrophage function (such as CD14, LGALS3, CAPG, INPPL1 and MAST2) were dysregulated in aging muscle, with skewing towards a pro-inflammatory phenotype. For example, the overrepresentation with aging of proteins such as Monocyte differentiation antigen CD14 (CD14, β = 0.003, p=0.009), Interferon-induced, double-stranded RNA-activated protein kinase (E2AK2, β = 0.0008 p=0.046) and ASC (PYCARD) (β = 0.006, p=0.025) can be viewed as being proinflammatory via their proposed role in lipid sensing and NF-κB activation (*Figure 6*B.1) (*Bryan et al., 2009*; *Sarkar et al., 2006*). Interestingly, we also identified proteins that were concurrently downregulated, such as Microtubule-associated serine/threonine-protein kinase 2 (MAST2, β = −0.002, p=0.023) and Phosphatidylinositol 3,4,5-trisphosphate 5-phosphatase 2 (INPPL1, β = −0.0009, p=0.036), that could accentuate the inflammatory phenotype by attenuating the negative regulation of NF-κB (*Figure 6*B.2) (*Kalesnikoff et al., 2002*; *Tridandapani et al., 2002*). Thus, increased expression of NF-κB activators and decreased expression of NF-κB attenuators may synergize to elevate chronic inflammation in aging muscle. We also noted increased expression of high mobility group protein B2 (HMGB2, β = 0.004, p=0.001), a well-known 'alarmin' (*Taniguchi et al., 2018*) that is released from dying cells or within neutrophil extracellular traps (NETs), that may further exacerbate the inflammatory milieu. Cumulatively, our observations are consistent with the enrichment of macrophages accumulation in aging muscle. Of note, a number of epidemiological studies have found that chronic inflammation is a risk factor for the development of sarcopenia, while the development of a proinflammatory state in adult mouse appears to interfere with tissue maintenance and repair, as evidenced by the fact that pharmacological inhibition of Jak2 and Stat3 activities stimulate the expansion of satellite cells in culture and their engraftment in vivo (*Costamagna et al., 2015*; *Price et al., 2014*; *Roth et al., 2006*). Interestingly, caloric restriction, one of the most effective interventions to counteracts aging in animal models, is associated with reduced inflammation in human muscle, as well as the reversal of some of the other age-related changes identified in this study, such as the increase in molecular chaperones (*Yang et al., 2016*).

Second, we found evidence of an anti-inflammatory activity that could potentially offset the proinflammatory milieu of aging muscle (*Figure 6C*). This was most evident in strong age-associated

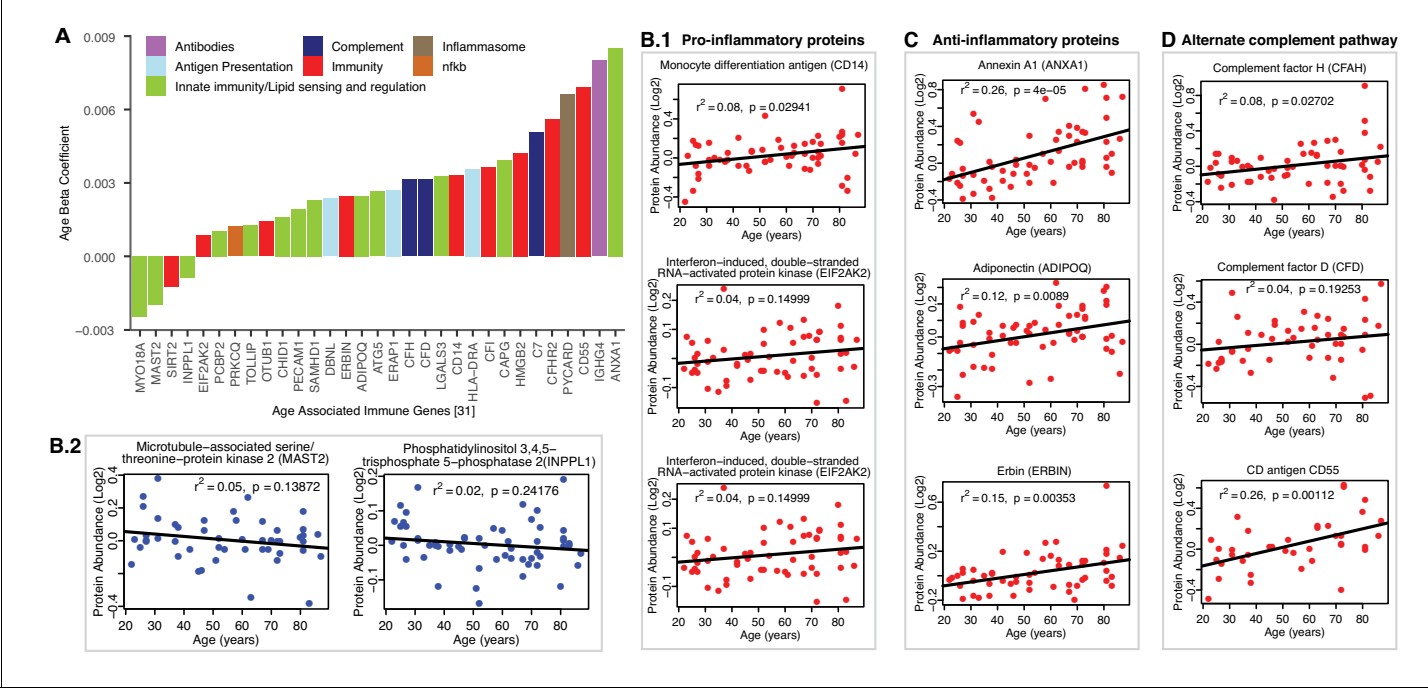

**Figure 6.** Age-associated immune proteins. (**A**) Immune-related proteins are depicted; the x axis shows the genes that code for age-differentially regulated proteins, while the y axis shows the log2 fold expression difference associated with age. The increase in innate immune signaling and lipid responses may indicate a reaction to adipocytes muscle infiltration, which in turn causes activation of innate immune signaling. (**B–D**) Examples of dysregulated proteins are shown from (B.1-B.4) pro-inflammatory. (**C**) Anti-Inflammatory. (**D**) Complement pathway proteins. Inflammasome adaptor protein PYCARD is positively associated with age, and the abundance of this protein is a key mechanism by which IL-1β pathway activation is regulated. In B-D raw log2 protein abundance and unadjusted p-values are shown.

DOI: https://doi.org/10.7554/eLife.49874.023

upregulation of annexin A1 (ANXA1, β = 0.008, p=0.00001), a protein that has been linked to resolution of inflammation (*Gobbetti and Cooray, 2016*). Elevated levels of adiponectin (ADIPOQ, β = 0.002, p=0.008), a chemokine produced exclusively by adipocytes, likely reflected increased adipogenic activity in aging muscle. However, it is interesting to note that ADIPOQ has also been proposed to inhibit endothelial NF-κB activation (*Ehsan et al., 2016*; *Chen et al., 2017*) and may, thereby, have context-dependent anti-inflammatory functions. Finally, erbin (ERBIN, β = 0.002, p=0.019), a nuclear lamina-associated protein that was overrepresented with age in our studies, has been implicated in reducing NF-κB activation by some stimuli (*McDonald et al., 2005*), with associated reduction in pro-inflammatory gene expression.

Third, coordinate upregulation of several members of the alternate complement pathway, such as CFAH (β = 0.003, p=0.028), and CFAD (β = 0.003, p=0.039) and modulators of complement activity such as CD antigen CD55 (DAF and CD55, β = 0.006, p=0.00007) indicate ongoing innate immune activity in aging muscle (*Figure 6D*). Whether this trend reflects increased presence of dying cells and cell debris or below-threshold autoimmune activity remains to be determined. The latter could be mediated, for example, by Immunoglobulin heavy constant gamma 4 (IGHG4, β = 0.008, p=0.019) which we found to be increased with age. This antibody isotype has been implicated in the generation of autoantibodies against muscle-specific kinases that are prevalent in certain forms of myasthenia gravis (*Hoch et al., 2001*). The possible connection between the aging muscle and chronic neurodegenerative disorders in which destruction of self-tissue by complement has been ascribed a causative role (*McGeer et al., 2017*) is an intriguing area for future investigation.

## Conclusions

The biological mechanisms that mediate the deleterious effect of aging on skeletal muscle are still controversial, as some evidence suggested that the decline of mitochondrial content, volume and

energetic efficiency plays a primary role, while other evidence showed show no significant change for the same parameters with aging, especially if the level of physical activity was considered (*Distefano and Goodpaster, 2018*). To investigate systematically the changes in expressed proteins that might drive the decline in skeletal muscle function, we conducted an in-depth quantitative measurement of age-related changes in protein abundance in human skeletal muscle. While we did not use model systems or in vivo experiments, because of the careful design of the study, the selection of a healthy population, the depth of protein detection and rigorous analysis made it possible to produce a descriptive quantitative dataset to show aging-associated molecular changes. We used a MS-based isobaric relative quantitative approach for proteome analysis that provides broad coverage of the proteins of human skeletal muscle in very healthy individuals over a wide age range and we adjusted our analysis for potential confounders. The biological function of most of the proteins reported in this study was gathered by an extensive review of the literature and instead of relying only on annotation of Uniprot or GO database, we manually curated the functional classification used in the analysis. We present evidence that our approach is robust and sensitive to true biological variability. We confirmed the altered expression of proteins implicated in pathways differentially active in human skeletal muscle with aging, including more highly abundant mitochondrial proteins and less abundant inflammatory proteins. We also identified subsets of proteins increasing with age that were not previously described, namely proteins implicated in alternative splicing and autophagy. An important limitation of this work is that proteomic analysis provides a static image of the protein concentration at one point in time and does not typically inform on the dynamics of protein accumulation or protein loss, the subcellular localization of proteins, or their post-translational modification (e.g. phosphorylation, ubiquitylation). Addressing these important parameters will help us interpret more fully the biological meaning of our findings and will be addressed as these studies progress (*Miller et al., 2019*). Our work complements a great deal of evidence from animal models that important metabolic and regulatory changes occur with aging in skeletal muscle, provides a rich resource to study the effect of aging on skeletal muscle proteome in humans and sets the stage for future research on the mechanisms driving the age-associated decline in muscle function.

## Materials and methods

### Study design and participants

Muscle biopsies analyzed in this study were collected from participants from the Genetic and Epigenetic Study of Aging and Laboratory Testing (GESTALT). Participants were enrolled in GESTALT if they were free of major diseases, except for controlled hypertension or a history of cancer that had been clinically silent for at least 10 years, were not chronically on medications (except one antihypertensive drug), had no physical or cognitive impairments, had a BMI less than 30 kg/m$^2$, and did not train professionally. Inclusion criteria were gathered from information on medical history, physical exams, and blood test interpreted by a trained nurse practitioner (*Schrack et al., 2014*). Participants were evaluated at the Clinical Research Unit of the National Institute on Aging Intramural Research Program. Data and muscle specimens from 60 participants were available for this study. However, two participants were excluded because the muscle specimen provided was too small to obtain reliable proteomic data. Therefore, data from 58 participants dispersed over a wide age-range (20–34 y, n = 13; 35–49 y, n = 11; 50–64 y, n = 12; 65–79 y, n = 12; 80+ y, n = 10) were used for this study. Anthropometric parameters were objectively assessed. The level of physical activity was determined using an interview-administered standardized questionnaire originally developed for the Health, Aging and Body Composition Study (*Brach et al., 2004*) and modeled after the Leisure-Time Physical Activity questionnaire (*Taylor et al., 1978*). Total participation time in moderate to vigorous physical activity per week was calculated by multiplying frequency by amount of time performed for each activity, summing all of the activities, then dividing by two to derive minutes of moderate to vigorous physical activity per week, the following categories were used: <30 min per week of high intensity physical activity was considered 'not active' and coded as 0; high-intensity physical activity $\geq$30 and<75 min was considered 'moderately active' and coded as 1, high-intensity physical activity $\geq$75 and<150 min was considered 'active' and coded as 2, and high-intensity physical activity $\geq$150 min was considered to 'highly active' and coded as 3. An ordinal variable from 0 to 3 was used in the analysis. The GESTALT protocol is approved by the Intramural Research Program of

the US National Institute on Aging and the Institutional Review Board of the National Institute of Environmental Health Sciences. All participants provided written, informed consent at every visit.

## Muscle biopsies

The depth of the subcutaneous fat (uncompressed and compressed) was determined using MRI images of the middle thigh performed on the previous day. A region above the vastus lateralis muscle was marked at the mid-point of a line drawn between the great trochanter and the mid-patella upper margin. The skin was prepped with povidone–iodine (Betadine) and ethyl alcohol, and the outside areas covered with sterile drapes. The biopsy site was anesthetized intradermally using a 27-gauge needle and then subcutaneously using a 23-gauge x 1 1/2 -inch needle, follow by an 18-gauge spinal needle, with ~15 mL of 1% lidocaine with sodium bicarbonate. The operator was careful that the anesthetic was infiltrated in the subcutaneous tissue and above the muscle fascia but not the muscle fibers not to distort the tissue structure and induce a gene expression response. A 6-mm Bergstrom biopsy needle was inserted through the skin and fascia incision into the muscle, and muscle tissue samples were obtained using a standard method. Biopsy specimens cut into small sections were snap frozen in liquid nitrogen and subsequently stored at −80°C until used for analyses.

## Sample preparation and protein extraction

On average 8 mg of muscle tissue was pulverized in liquid nitrogen and mixed with the lysis buffer containing protease inhibitor cocktail (8 M Urea, 2M Thiourea, 4% CHAPS, 1% Triton X-100, 50 mM Tris, pH 8.5 (Sigma)). Protein concentration was determined using commercially available 2-D quant kit (GE Healthcare Life Sciences). Sample quality was confirmed using NuPAGE protein gels stained with fluorescent SyproRuby protein stain (Thermo Fisher).

In order to remove detergents and lipids 300 µg of muscle tissue lysate were precipitated using standard methanol/chloroform extraction protocol (*Bligh and Dyer, 1959*). Proteins were resuspended in concentrated urea buffer (8 M Urea, 2 M Thiourea, 150 mM NaCl (Sigma)), reduced with 50 mM DTT for 1 hr at 36°C and alkylated with 100 mM iodoacetamide for 1 hr at 36°C in the dark. The concentrated urea was diluted 12 times with 50 mM ammonium bicarbonate buffer and proteins were digested for 18 hr at 36°C using trypsin/LysC mixture (Promega) in 1:50 (w/w) enzyme to protein ratio. Protein digests were desalted on 10 × 4.0 mm C18 cartridge (Restek, cat# 917450210) using Agilent 1260 Bio-inert HPLC system with the fraction collector. Purified peptides were speed vacuum-dried and stored at −80°C until further processing.

Tandem Mass Tags (TMT) labeling was used to perform quantitative proteomics. Each TMT labeling reaction contains six labels to be multiplexed in a single MS run. Donor IDs were blinded, and samples were randomized to prevent TMT bias. Each TMT 6-plex set included one donor from each of the five age groups and one reference sample. 5 muscle samples 100 µg each corresponding to five different age groups and one separately prepared master reference sample were labeled with 6-plex tandem mass spectrometry tags using a standard TMT labeling protocol (Thermo Fisher). 200 femtomole of bacterial beta-galactosidase digest (SCIEX) was spiked into each sample prior to TMT labeling to control for labeling efficiency and overall instrument performance. Labeled peptides from six different TMT channels were combined into one experiment and fractionated.

## High-pH RPLC fractionation and concatenation strategy

High-pH RPLC fractionation was performed on Agilent 1260 bio-inert HPLC system using 3.9 mm X 5 mm XBridge BEH Shield RP18 XP VanGuard cartridge and 4.6 mm X 250 mm XBridge Peptide BEH C18 column (Waters). Solvent composition was as follows: 10 mM ammonium formate (pH 10) as mobile phase (A) and 10 mM ammonium formate and 90% ACN (pH 10) as mobile-phase B (*Wang et al., 2011*).

TMT-labeled peptides prepared from the skeletal muscle tissues were separated using a linear organic gradient that went from 5% to 50% B in 100 min. Initially, 99 fractions were collected during each LC run at 1 min interval each. Three individual high-pH fractions were concatenated into 33 combined fractions with the 33 min interval between each fraction (fractions 1, 34, 67 = combined fraction 1, fractions 2, 35, 68 = combined fraction two and so on). Combined fractions were speed vacuum dried, desalted and stored at −80°C until final LC-MS/MS analysis.

## LC-MS/MS analyses

Purified peptide fractions from skeletal muscle tissues were analyzed using UltiMate 3000 Nano LC Systems coupled to the Q Exactive HF mass spectrometer (Thermo Scientific, San Jose, CA). Each fraction was separated on a 35 cm capillary column (3 µm C18 silica, Hamilton, HxSil cat# 79139) with 150 um ID on a linear organic gradient using 650 nl/min flow rate. Gradient went from 5% to 35% B in 205 min. Mobile phases A and B consisted of 0.1% formic acid in water and 0.1% formic acid in acetonitrile, respectively. Tandem mass spectra were obtained using Q Exactive HF mass spectrometer with the heated capillary temperature +280°C and spray voltage set to 2.5 kV. Full MS1 spectra were acquired from 300 to 1500 m/z at 120,000 resolution and 50 ms maximum accumulation time with automatic gain control [AGC] set to $3 \times 10^6$. Dd-MS2 spectra were acquired using dynamic m/z range with fixed first mass of 100 m/z. MS/MS spectra were resolved to 30,000 with 155 ms of maximum accumulation time and AGC target set to $2 \times 105$. Twelve most abundant ions were selected for fragmentation using 30% normalized high collision energy. A dynamic exclusion time of 40 s was used to discriminate against the previously analyzed ions.

## Proteomics informatics

The mgf files generated (using MSConvert, ProteoWizard 3.0.6002) from the raw data from each sample fraction was searched with Mascot 2.4.1 and X!Tandem CYCLONE (2010.12.01.1) using the SwissProt Human sequences from Uniprot (Version Year 2015, 20,200 sequences, appended with 115 contaminants) database. The search engine was set with the following search parameters: TMT6plex lysine and n-terminus as fixed modifications and variable modifications of carbamido-methyl cysteine, deamidation of asparagine and glutamate, carbamylation of lysine and n-terminus and oxidized methionine. A peptide mass tolerance of 20 ppm and 0.08 Da, respectively, and two missed cleavages were allowed for precursor and fragment ions in agreement with the known mass accuracy of the instrument. Mascot and X!Tandem search engine results were analyzed in Scaffold Q + 4.4.6 (Proteome Software, Inc, RRID:SCR_014345). The TMT channels' isotopic purity was corrected according to the TMT kit.peptide and protein probability was calculated by PeptideProphet (*Keller et al., 2002*) and ProteinProphet probability model (*Nesvizhskii et al., 2003*). The Peptide-Prophet model fits the peptide-spectrum matches into two distributions, one an extreme value distribution for the incorrect matches, and the other a normal distribution for correct matches. The protein was filtered at thresholds of 0.01% peptide FDR, 1% protein FDR and requiring a minimum of 1 unique peptide for protein identification. We allow single peptide hits for two reasons: first, any peptide that is quantifiable is detected at least across 25% of all samples (n = 58); second, we identify proteins with more than one search engine, so the protein identification is confirmed at least twice, even for single-peptide hits. For these reasons, the even single peptides are unlikely to be random hits. As for single peptide quantification, the spectrum-to-spectrum variability is no higher between spectra from the same peptide than between spectra from different peptides from the same protein. Therefore, it is unlikely that there is any differential 'bias' in reporter ions from peptide to peptide. More importantly, TMT is taken as relative, not absolute, quantification. So even if there were such a bias, it would be the same across samples, so the relative quantification would not be affected. Reporter ion quantitative values were extracted from Scaffold and decoy spectra, contaminant spectra and peptide spectra shared between more than one protein were removed. Typically, spectra are shared between proteins if the two proteins share most of their sequence, usually for protein isoforms. Reporter ions were retained for further analyses if they were exclusive to only one protein, and they were identified in all six channels across each TMT set. Since we have multiple age group across each TMT experiment, we analyzed the proteins for missing reporter ion intensity.

For this analysis, the protein with missing reporter ion in some of the channels (not more than two channels) for each TMT experiment was identified and missing value imputation was performed using multiple imputation with chained equations (MICE) R library by predictive mean matching. Mean imputation was performed <0.01% in one or two TMT channels in most of the TMT experiments, except TMTset1 (the missing reporter ion for channel 5 is 0.03%). The reporter ion intensity from the proteins derived from the imputation method (on an average <10 proteins) were concatenated with reporter ion intensity identified in all six channels and further analysis performed using adjudicated values. The log2 transformed reporter ion abundance was normalized by median subtraction from all reporter ion intensity spectra belonging to a protein across all channels

(*Kammers et al., 2015*; *Herbrich et al., 2013*). Relative protein abundance was estimated by median of all peptides for a protein combined. Protein sample loading effects from sample preparations were corrected by median polishing, that is subtracting the channel median from the relative abundance estimate across all channels to have a median zero as described elsewhere (*Kammers et al., 2015*; *Herbrich et al., 2013*). Quantified proteins were clustered together if they shared common peptides and corresponding gene names were assigned to each protein for simplicity and data representation. Annotation of the proteins were performed by manual curation and combining information from Uniprot, GO and PANTHER database. Further bioinformatics analysis was performed using R programming language (3.4.0) and the free libraries available on Bioconductor.

## Linear mixed effect model and statistical analyses

Linear mixed regression model was implemented to examine age effects and the data was adjusted for physical activity, gender, race, bmi, type I and type II myosin fiber ratio and TMT mass spectrometry experiments. Protein significance from the regression model was determined with p-values derived from lmerTest. Partial Least Square analysis (PLS) was used to derive models with classification that maximized the variance between age groups. PLS loadings were derived from linear model adjusted protein results. The regression model was performed using R 3.3.4 (*R Development Core Team, 2016*) with lme4 v1.1. library. Heat maps and hierarchical cluster analysis were performed using the non-linear minimization package in R (*Gaujoux and Seoighe, 2010*). GraphPad PRISM 6.07 and R packages were used for statistical analysis and generation of figures. STRING analysis (10.5 version) was used for obtaining protein-protein interaction network. Enrichment analysis was performed by GeneSet Enrichment Analysis (GSEA) and PANTHER, the pathways were mapped and visualized by Cytoscape 3.0 (*Shannon et al., 2003*). One-way ANOVA, nonparametric, and chi-square tests (continuous and categorical variables) were used to test for sample differences between age groups.

## RNA extraction and purification

Total RNA was prepared by lysing cell pellets (2 × 106) in 700 µl Qiazol and extracted using Qiagen miRNeasy mini kit according to the manufacturer's recommendation (Qiagen Inc, CA) from the same samples (n = 54). Small ribosomal RNA was further depleted using Qiagen GeneRead rRNA Depletion Nano Kit. Total RNA quality and quantity was checked using RNA-6000 nano kits on the Agilent 2100-Bioanalyzer. 375 ng of high-quality RNA was used for first-strand and second-strand cDNA synthesis followed by single primer isothermal amplification (SPIA) using NuGEN Ovation RNA–Seq System V2 kits according to manufacturer's protocol. This kit amplified both polyA-tailed and non-polyA tailed RNA and removed ribosomal RNA. The amplified cDNA was sheared using Bioruptor (Diagenode) to an average size of 150–250 bases. The sequencing library was prepared using Illumina ChIP-Seq kits according to the manufacturer's protocol (Illumina, San Diego, CA). In short, the ends of the fragments were repaired using T4 DNA polymerase, *E. coli* DNA Pol I large fragment (Klenow polymerase), and T4 polynucleotide kinase (PNK) and an A-overhang was added to the 3' end. Adapters were ligated to the DNA fragments and size-selected (250–350 bases) on a 4.5% agarose gel. An 18-cycle PCR amplification was performed followed by a second 4.5% agarose gel size selection before cluster generation in cbot2 and sequencing with Illumina Hiseq2500 sequencer using V4 reagents. Single-read sequencing was performed for 138 cycles and Real-Time Analysis (RTA) v1.18.66.3 generated the base-call files (BCL files). BCL files were de-multiplexed and converted to standard FASTQ files using bcl2fastq program (v2.17.1.14).

## RNA-Seq quantification and splicing analysis

The quality of the bases was checked using FASTQC program (v11.2) before and after adapter removal and last base trimming by cutadapt program (v1.9). The cleaned FASTQ files were aligned, quantified and annotated to the human hg19 genome using Salmon (*Patro et al., 2017*) with the concept of quasi-mapping with two phase inference procedure for gene model annotations. The GC bias corrected, quantified transcript isoform abundance values (TPM) were used for further computation of relative abundance of the events or transcripts isoforms known as percent spliced-in (PSI) by SUPPA (*Alamancos et al., 2015*). Since the variability of low-expressed genes between biological

replicates were reported, the transcript data were filtered for the transcripts which were expressed in at least three donors per each age group. Thus, we excluded ~23% of the transcripts from total transcript quantification for further splicing analysis. Events coordinates are extracted from the Ensembl annotation (GRCh37.75) and alternative splicing events were generated. PSI values of alternative splicing events for each transcript from each sample (n = 53) were estimated and the PSI values showing a good agreement with the RNA seq data were kept for further analysis. The magnitude of the PSI change (differential splicing) across the age were calculated with a linear mixed model analysis performed on the PSI to estimate the age-related splicing changes of the transcript isoform. The PSI regression model was adjusted with the aging confounders as same as described above for protein regression model except fiber ratio. For transcript data, we used RNA experiment batches as a random effect. The age beta coefficient for each alternative splicing event transcript PSI was reported as the magnitude to the splicing event-specific PSI change with age. Significance of the alternative splicing events was calculated by lmerTest and was reported if the observation had a p-value<0.05 at transcript level for age beta coefficient.

## Age-association of proteins and transcripts

Proteins or transcripts either significantly upregulated or down regulated with age, present in 50% of the samples or at least in three samples for each age group, and significant (p<0.05) were considered as age-associated. Age-association was measured by linear mixed model adjusted for confounders of aging phenotype either in protein analysis or in RNAseq analysis and were further filtered for significance calculation. Age beta coefficient for each protein or transcript were calculated from log2 normalized data on which a mixed linear regression model was applied. Thus, the age beta coefficient represents the mean log2 fold expression per year of age. LmerTest was used for calculating p-values from t-tests via Satterthwaite's degrees of freedom method. Any protein or transcript was represented as age-associated if the p-value for the protein or transcript was <0.05. p-Values for multiple comparisons were adjusted using Benjamini-Hochberg method in R and adjusted p-values were reported on related tables (figure source data). Age-associated proteins and age-associated alternatively spliced transcripts were further analyzed into two categories, either age-association beta coefficient (<0) was under represented with age–indicating a decrease in the abundance of the protein with a year of age or age-association beta coefficient (>0) was over represented with age-indicating the abundance of the protein was increased with a year of age. For simplicity of reporting, we calculated the enrichment of these proteins/transcripts over the total age-associated protein/transcripts and reported as underrepresented and overrepresented with age.

## Data availability

The mass spectrometry proteomics data have been deposited to the ProteomeXchange Consortium via the PRIDE partner repository with the dataset identifier PXD011967 (*Ubaida-Mohien et al., 2019*). RNASeq data is deposited in GEO (GSE129643).

## Acknowledgements

This work was supported by the Intramural Research Program of the National Institute on Aging, NIH, Baltimore, MD, USA, NIH R01 AG027012, and R01 AG057723. We are grateful to the GESTALT participants and the GESTALT Study Team at Harbor Hospital and NIA, Linda Zukely, and Mary Kaila for sample collection and project coordination. We thank Supriyo De, Yulan Piao and the NIA Sequencing Core Facility for the RNA sample sequencing, library preparation and data generation. We also thank Lauren Brick for assistance with figure design.

## Additional information

### Funding

| Funder | Grant reference number | Author |
|--------|------------------------|--------|
| National Institute on Aging | Intramural Research Program | Ceereena Ubaida-Mohien<br>Alexey Lyashkov<br>Marta Gonzalez-Freire<br>Ravi Tharakan<br>Michelle Shardell<br>Ruin Moaddel<br>Chee W Chia<br>Myriam Gorospe<br>Ranjan Sen<br>Luigi Ferrucci |
| National Institutes of Health | NIH R01 AG027012 | Richard D Semba |
| National Institutes of Health | NIH R01 AG057723 | Richard D Semba |

The funders had no role in study design, data collection and interpretation, or the decision to submit the work for publication.

### Author contributions

Ceereena Ubaida-Mohien, Alexey Lyashkov, Marta Gonzalez-Freire, Ravi Tharakan, Michelle Shardell, Ruin Moaddel, Richard D Semba, Chee W Chia, Myriam Gorospe, Ranjan Sen, Formal analysis, Writing—review and editing, Conceptualization, Resources, Formal analysis, Supervision, Funding acquisition, Investigation, Project administration; Luigi Ferrucci, Conceptualization, Resources, Formal analysis, Supervision, Funding acquisition, Investigation, Project administration, Writing—review and editing

### Author ORCIDs

Luigi Ferrucci (iD) https://orcid.org/0000-0002-6273-1613

### Ethics

Human subjects: The GESTALT protocol is approved by the Intramural Research Program of the US National Institute on Aging and the Institutional Review Board of the National Institute of Environmental Health Sciences (Protocol Number: 15-AG-0063). All participants provided written, informed consent at every visit.

### Decision letter and Author response

Decision letter https://doi.org/10.7554/eLife.49874.029
Author response https://doi.org/10.7554/eLife.49874.030

## Additional files

### Data availability

The mass spectrometry proteomics data have been deposited to the ProteomeXchange Consortium via the PRIDE partner repository with the dataset identifier PXD011967. RNASeq data is deposited in GEO (GSE129643).

The following datasets were generated:

| Author(s) | Year | Dataset title | Dataset URL | Database and Identifier |
|-----------|------|---------------|-------------|-------------------------|
| Ceereena Ubaida-Mohien, Luigi Ferrucci | 2019 | Proteomics of Human Skeletal Muscle | http://proteomecentral.proteomexchange.org/cgi/GetDataset?ID=PXD011967 | ProteomeXchange, PXD011967 |

| Ceereena Ubaida-Mohien, Luigi Ferrucci | 2019 | Skeletal Muscle Transcriptomics | https://www.ncbi.nlm.nih.gov/geo/query/acc.cgi?acc=GSE129643 | NCBI GeneExpression Omnibus, GSE129643 |

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
