## [Decision Letter]

Thank you for submitting your article "Aging skeletal muscle proteomics finds changes in spliceosome, immune factors, proteostasis and mitochondria" for consideration by *eLife*. Your article has been reviewed by three peer reviewers, including Matt Kaeberlein as the Reviewing Editor and Reviewer #1, and the evaluation has been overseen by Clifford Rosen as the Senior Editor. The following individual involved in review of your submission has agreed to reveal their identity: Rozalyn Anderson (Reviewer #2).

The reviewers have discussed the reviews with one another and the Reviewing Editor has drafted this decision to help you prepare a revised submission. Overall there was marked enthusiasm for this paper, which should provide a resource for further investigators. The revisions should be relatively easy to accomplish.

Reviewer #1:

This is an important and interesting analysis of muscle proteomics from healthy people aged between 20 to 87 years old. This provides a fantastic resource to the community that will spur additional studies, and the authors also identified some key changes in metabolic, immune-related, splicing, and proteostasis-related proteins with age. Although it is difficult to determine causality from this type of study, there is a dearth of large-scale, high-quality human studies in this area, and this represents a significant step forward.

1) I'm not sure the phrase "extremely healthy" in the Introduction is appropriate. Perhaps just "healthy".

2) Again in the conclusion the phrase "extraordinary healthy" is used. I feel that this needs to be quantified or rigorously defined. In what way are they "extraordinary"? How was this measured? Otherwise, just "healthy" seems more appropriate.

Reviewer #2:

This is a very interesting study from the NIA intramural program. The GESTALT study is designed specifically to uncover the biology of human aging. The strength of this study is that biometric, clinical, and functional measures are captured for the individuals enrolled and these data can be used to interrogate molecule profiling data from collected tissue specimens. In this report, the team describe molecular studies from skeletal muscle biopsies from individuals representing age groups across adult lifespan. The team finds signatures of human aging that are entirely consistent with what might be expected from work in shorter lived species, namely metabolism and immune/inflammatory pathways. This is important because it underscores the translatability of insights from other species in which mechanistic studies might be undertaken. A novel and intriguing finding of this study is that RNA processing seems to be engaged as part of the adaptation to age, this new insight is evident first in the proteomic data and confirmed nicely in the RNAseq data showing numerous examples of splicing events altered as a function of age. The paper is beautifully written, the data are nicely presented and carefully explained. The statistical methods using beta coefficients to track the impact of aging on individual parameters is very attractive. Using this approach, the authors have identified a credible and exciting narrative of human skeletal muscle aging at unprecedented resolution.

The strength of this study is its potential to uncover those mechanistic insights generated in short lived species that are directly relevant to human aging and thereby highly translational. While it is appreciated how much work was done scouring the literature to identify potential roles of the identified proteins, there is a middle ground missing and the broader context is a bit lost in the details of individual factors. The authors have not been as thorough in their coverage of what is already known from common laboratory animals. It would strengthen the paper to provide a little more context from (a) rodent and nonhuman primate muscle aging and (b) more generally, pathways implicated in aging not just in human disease.

The focus on metabolism is really great and there is a literature on mitochondrial decline with age in humans and in other species. There are a few papers that might be of interest to the authors; monkey skeletal muscle aging is associated with the same energetic and redox changes that the authors predict here from their proteomics data (Pugh et al., 2013), there are also several aging and exercise papers that have pointed to this same signature (Melov et al., 2007, PLoS One; Radom-Aizik et al., 2005, Med. Sci. Sports Exerc.), and another that shows functional mitochondrial decline associated with frailty (Andreux et al., 2018, Sci. Rep.). The authors may be interested to know that the signature of aging identified here is the inverse of the signature of CR identified across tissues and among species (Barger et al., 2015, PLoS One).

There is an interesting change in pace of aging after 50 years of age, this is also the time in life where energetics and the impact of physical activity on metabolic profiles changes (Speakman and Westerterp, 2010) – the links between whole body metabolic function and physical activity may well originate from age-related changes in skeletal muscle.

The proteostasis data are very interesting and point to a change in the balance of predominant turnover mechanisms including autophagy, proteasome, chaparones. There are a few papers/reviews from the non-human literature that might be of interest (Altun et al., 2010, J. Biol. Chem; Martinez et al., 2017, Aging Cell; Sands et al., 2017). These are only a few examples to showcase what might be considered in a broader approach in providing the context for the current study.

In the section on ribosomal proteins the authors might consider some of the work by Hamilton and Miller at Colorado State, there are data on protein turnover specifically in skeletal muscle as a function of age, although not in humans.

RNA-based mechanisms are an emerging topic in biomedical research, and recent studies have identified a role for RNA splicing in mechanisms of CR (Heintz et al., 2017, Nature; Rhoads et al., 2018, Cell Metab.). The splicing story is very exciting and one that will be of considerable interest to the readership.

The data shown in Figure 1 is really about technical aspects of reproducibility and quality control and would be better suited to the supplement.

Reviewer #3:

The manuscript titled "Aging Skeletal Muscle Proteomics Finds Changes in Spliceosome, Immune factors, Proteostasis and Mitochondria" by Ubaida-Mohien is an excellent work performing a very large study from muscle biopsies from the GESTALT cohort. 58 healthy individuals were enlisted from this precious cohort to assess changes with aging. This study has provided a wealth of data on a very well designed experiment – very deep coverage into the muscle proteome was achieved with 5,891 unique proteins reported. A lot of data was acquired (12 TMT plex were fractionated off-line and finally for each TMT plex 33 MS acquisitions were acquired). The conclusions that the authors report are also very interesting in that 29% of the muscle proteome changes with aging. The significantly changing proteins were subsequently also correlated to aging scores. The authors are particularly pointing out changes in splicing-related proteins, which is very interesting. This aspect of age-associated alternative splicing and splicing events was further followed up with transcriptomics experiments (RNAseq of 53 of these individuals). Overall this is excellent work which greatly contributes to our understanding of muscle aging.

Some technical aspects should be addressed in data presentation.

- The authors mention under data availability "The mass spectrometry proteomics data have been deposited to the ProteomeXchange Consortium via the PRIDE partner repository with the dataset identifier PXD011967". In the repository, it indicated the data set was referring to another publication also – could the authors please comment or also else cite that manuscript, if appropriate.

- It would be important to provide a list that matches the mass spectrometric raw files (and TMT channels) with the sample code name for the individual subjects within the TMT, or at least their age classification. Currently, the raw data is not interpretable to other researchers as it is not clear what TMT channels are derived from which age groups. The TMT labeling strategies should be listed and annotated in a supplementary file for each of the 12 TMT6plex experiments that the authors list. That way this very rich and very valuable data set can also be reanalyzed by other researchers.

- The authors are thorough in providing all quantitative information from the TMT quantification from Figure 1—source data 1. TMT experiments are numbers TMT1 through TMT 12 and channels are listed – similar to the above could the authors also indicate the age group annotation in this list.

- While the quantitative data and also statistical processing thereof is provided the authors should also provide a supplementary table that lists all protein and peptide identification data that is processed and filtered for confident peptides. The authors provide uploaded data files (as search engine results). But it will be important to know what (and how many) peptides were identified per each protein – and listing their peptide precursor ions, scores. Do protein assignments conform with the requirement of 2 observed peptides per protein? This can currently not be assessed.

---

## [Author Response]

Reviewer #1:[…]1) I'm not sure the phrase "extremely healthy" in the Introduction is appropriate. Perhaps just "healthy".2) Again in the conclusion the phrase "extraordinary healthy" is used. I feel that this needs to be quantified or rigorously defined. In what way are they "extraordinary"? How was this measured? Otherwise, just "healthy" seems more appropriate.

We agree that the expression “extremely healthy” may be too strong and we have removed it. We want to point out that the condition of “healthy” in GESTALT is not a subjective judgement but was rather based on a comprehensive set of pre-defined criteria that were objectively assessed during a screening visit in the NIA clinic by expert medical professionals. This approach is substantially more rigorous than what has been used by many other studies that defined “healthy” subjects based on a summary medical history. The main criteria included:

1) Absence of any chronic disease, with the exception of hypertension maintained under control (<140/80 mmHg) with one single drugs, and cancer that had been clinically silent for at least 10 years;

2) No chronic drug treatment (with the exception of the one antihypertensive cited above);

3) Ability to walk 400 m without developing symptoms such as shortness of breath;

4) No cognitive impairment evaluated through objective cognitive testing, and;

5) No overt abnormal values to a series of blood tests performed during the screening visits, which included both clinical chemistry and CBC with differential count.

Thus, when we mentioned “extremely healthy” we wanted to point out the we arrive at this definition through in-depth clinical examination lasting 4-6 hours, performed by expert nurse practitioners, and reviewed during a clinical conference by the entire study team.

Reviewer #2:[…] The strength of this study is its potential to uncover those mechanistic insights generated in short lived species that are directly relevant to human aging and thereby highly translational. While it is appreciated how much work was done scouring the literature to identify potential roles of the identified proteins, there is a middle ground missing and the broader context is a bit lost in the details of individual factors. The authors have not been as thorough in their coverage of what is already known from common laboratory animals. It would strengthen the paper to provide a little more context from (a) rodent and nonhuman primate muscle aging and (b) more generally, pathways implicated in aging not just in human disease.The focus on metabolism is really great and there is a literature on mitochondrial decline with age in humans and in other species. There are a few papers that might be of interest to the authors; monkey skeletal muscle aging is associated with the same energetic and redox changes that the authors predict here from their proteomics data (Pugh et al., 2013), there are also several aging and exercise papers that have pointed to this same signature (Melov et al., 2007, PLoS One; Radom-Aizik et al., 2005, Med. Sci. Sports Exerc.), and another that shows functional mitochondrial decline associated with frailty (Andreux et al., 2018, Sci. Rep.). The authors may be interested to know that the signature of aging identified here is the inverse of the signature of CR identified across tissues and among species (Barger et al., 2015, PLoS One).

We appreciate this suggestion. In an attempt to make sense of the massive complexity of the data produced by this study, we may have provided too many details. In the revised manuscript, we added a brief overall interpretation of our findings at the end of each section. We also included a brief comparison with data that had been published from studies in animal models. In these sections, we reference papers suggested by the reviewers, and other articles that we found while searching the literature. We limited the comparison to just a few examples, because an exhaustive review of the literature would have required lengthening too much this already massive manuscript. Larger sections of text were included at: subsection “Focus on the Aging Biological Mechanisms”, fifth paragraph; subsection “Decline of Mitochondrial Proteins with Age”, last paragraph; subsection “Pro-inflammatory and Anti-Inflammatory Immune Proteins of Aging Muscle”, first paragraph; subsection “Conclusions”.

There is an interesting change in pace of aging after 50 years of age, this is also the time in life where energetics and the impact of physical activity on metabolic profiles changes (Speakman and Westerterp, 2010) – the links between whole body metabolic function and physical activity may well originate from age-related changes in skeletal muscle.

We appreciate the reviewers bringing to our attention the article by Speakman and Westerterp. Indeed, it is quite interesting that the data reported by the authors are so consistent with the shift in protein composition in skeletal muscle that we detect in our analysis. We now reference this paper to underscore the idea that there is a dynamic relationship between change in body composition and changes in energy metabolism, although it is still unclear which of these changes happens first (subsection “Decline of Mitochondrial Proteins with Age”, last paragraph).

The proteostasis data are very interesting and point to a change in the balance of predominant turnover mechanisms including autophagy, proteasome, chaparones. There are a few papers/reviews from the non-human literature that might be of interest (Altun et al., 2010, J. Biol. Chem; Martinez et al., 2017, Aging Cell; Sands et al., 2017). These are only a few examples to showcase what might be considered in a broader approach in providing the context for the current study.

We agree that the data on proteostasis are interesting, particularly as they support the notions that proteins contributing to chaperone-mediated proteostasis appear to be underrepresented while those involved in macroautophagy appear overrepresented, possibly as a compensatory mechanism. We now cite literature supporting the notion that autophagy declines with aging in multiple organisms, including data from the comparative biology perspective, and suggesting that such a decline may be one of the intrinsic mechanisms of aging (see subsection “Differential Regulation of Proteins Related to Proteostasis in Aging”, last paragraph).

In the section on ribosomal proteins the authors might consider some of the work by Hamilton and Miller at Colorado State, there are data on protein turnover specifically in skeletal muscle as a function of age, although not in humans.

We thank the reviewer for this great suggestion. Proteomic analysis is powerful, but only provides a snapshot of the proteins present in one sample at one point in time. Of course, the concentration of that protein is the result of a tightly regulated equilibrium between synthesis and degradation. Different proteins have different turnover rates and considering this information would be important for a more nuanced interpretation of the proteomic data. We are currently in touch with Benjamin Miller at Oklahoma Medical Research Foundation and plan to include protein turnover in our analytical modelling as our studies progress. A comment concerning this problem was added (subsection “Conclusions”).

RNA-based mechanisms are an emerging topic in biomedical research, and recent studies have identified a role for RNA splicing in mechanisms of CR (Heintz et al., 2017, Nature; Rhoads et al., 2018, Cell Metab.). The splicing story is very exciting and one that will be of considerable interest to the readership.

Yes, we are quite excited about the RNA splicing story. Our earlier epidemiological studies revealed that mRNAs encoding splicing factors change with aging (Lee et al., 2019, Biogerontology; Lee et al., 2016, Aging Cell; Holly et al., 2013; Holly et al., 2014, Mech. Ageing Dev.; Harries et al., 2011). Since the results in the current manuscript confirmed the finding that components of the splicing machinery factors change with age, we intend to work in this specific area of research using high-depth RNA sequencing, and long-RNA analysis using the Nanopore technology. As mentioned in the Discussion section, alternative splicing may be a primary mechanism to increase resiliency, allowing cells to offset some of the damaging effects of aging, especially in the area of energetics (subsection “Age-Associated Alternative Splicing and Splicing Events”, last paragraph).

The data shown in Figure 1 is really about technical aspects of reproducibility and quality control and would be better suited to the supplement.

We agree with this suggestion. Figure 1 has now been moved to Figure 1—figure supplement 1.

Reviewer #3:[…]- The authors mention under data availability "The mass spectrometry proteomics data have been deposited to the ProteomeXchange Consortium via the PRIDE partner repository with the dataset identifier PXD011967". In the repository, there it indicated the data set was referring to another publication also – could the authors please comment or also else cite that manuscript, if appropriate.- It would be important to provide a list that matches the mass spectrometric raw files (and TMT channels) with the sample code name for the individual subjects within the TMT, or at least their age classification. Currently, the raw data is not interpretable to other researchers as it is not clear what TMT channels are derived from which age groups. The TMT labeling strategies should be listed and annotated in a supplementary file for each of the 12 TMT6plex experiments that the authors list. That way this very rich and very valuable data set can also be reanalyzed by other researchers.

We appreciate this suggestion and have included a document in the EBI PRIDE repository with detailed instructions for decoding the sample files. Also included a source data file (Figure 1—source data 1) with manuscript for sample annotation details and TMT labeling strategies. The manuscript to which the PRIDE dataset refers was a different analysis of the same dataset, used to assess how skeletal muscle was affected by physical activity, instead of age as we have done here. The paper has been cited in the manuscript (subsection “Decline of Mitochondrial Proteins with Age”, last paragraph) and cited on in the Materials and methods section (subsection “Data availability”).

- The authors are thorough in providing all quantitative information from the TMT quantification from Figure 1—source data 1. TMT experiments are numbers TMT1 through TMT 12 and channels are listed – similar to the above could the authors also indicate the age group annotation in this list.

We thank the reviewer for this request. The new Figure 1—source data 1 includes the age and gender for each TMT channel.

- While the quantitative data and also statistical processing thereof is provided the authors should also provide a supplementary table that lists all protein and peptide identification data that is processed and filtered for confident peptides. The authors provide uploaded data files (as search engine results). But it will be important to know what (and how many) peptides were identified per each protein – and listing their peptide precursor ions, scores. Do protein assignments conform with the requirement of 2 observed peptides per protein? This can currently not be assessed.

We appreciate this request as well. The number of peptides per protein is reported in the new table (Figure 1—source data 1). We used both Mascot and X!Tandem for peptide and protein identification, and PeptideProphet for post-processing. Thus, we can report the final PeptideProphet analysis of the Mascot and X!Tandem score for each peptide (Figure 1—source data 1; 24 additional excel sheets for 12 TMT experiments) and protein identification probability score for proteins (Figure 1—source data 1). The source dataset is described in Figure 1—source data 1, and the method has been specified in the Materials and methods section (subsection “Proteomics informatics”), and as we state there, we allow one peptide per protein matches, based on the work of Gupta and Pevzner (2009, J. Proteome Res.).